# Frizzled-8 integrates Wnt-11 and transforming growth factor-β signaling in prostate cancer

Virginia Murillo-Garzón[1], Irantzu Gorroño-Etxebarria[1], Malin Åkerfelt[2], Mikael Christer Puustinen[3], Lea Sistonen[3], Matthias Nees[2], James Carton[2], Jonathan Waxman[4] & Robert M. Kypta [1,4]

Wnt-11 promotes cancer cell migration and invasion independently of β-catenin but the receptors involved remain unknown. Here, we provide evidence that $FZD_8$ is a major Wnt-11 receptor in prostate cancer that integrates Wnt-11 and TGF-β signals to promote EMT. FZD8 mRNA is upregulated in multiple prostate cancer datasets and in metastatic cancer cell lines in vitro and in vivo. Analysis of patient samples reveals increased levels of $FZD_8$ in cancer, correlating with Wnt-11. $FZD_8$ co-localizes and co-immunoprecipitates with Wnt-11 and potentiates Wnt-11 activation of ATF2-dependent transcription. FZD8 silencing reduces prostate cancer cell migration, invasion, three-dimensional (3D) organotypic cell growth, expression of EMT-related genes, and TGF-β/Smad-dependent signaling. Mechanistically, $FZD_8$ forms a TGF-β-regulated complex with TGF-β receptors that is mediated by the extracellular domains of $FZD_8$ and TGFBR1. Targeting $FZD_8$ may therefore inhibit aberrant activation of both Wnt and TGF-β signals in prostate cancer.

[1] Center for Cooperative Research in Biosciences, CIC bioGUNE, 48160 Derio, Spain. [2] Institute of Biomedicine, University of Turku, FI-20520 Turku, Finland. [3] Faculty of Science and Engineering, Åbo Akademi University, FI-20014 Turku, Finland. [4] Department of Surgery and Cancer, Imperial College London, London W12 0NN, UK. Correspondence and requests for materials should be addressed to R.M.K. (email: rmkypta@gmail.com)

Prostate cancer is the most commonly diagnosed cancer and the second leading cause of death in men in Western countries[1]. Owing to the essential role of the androgen receptor (AR) in the normal growth and development of the prostate gland, and also in prostate carcinogenesis[2], men with prostate tumors initially respond well to androgen deprivation therapy[3]. However, most patients eventually experience disease progression to a more aggressive state, defined as castration-resistant prostate cancer (CRPC)[4]. Although a new generation of drugs that target AR signaling is extending the lives of patients with CRPC[4,5], the development of treatment resistance remains an issue. Consequently, the identification of targets not involving AR could lead to the development of more effective treatments.

Wnt proteins are a family of cysteine-rich secreted lipoglyco-proteins that play fundamental roles in development and disease[6]. Dysregulation of Wnt signaling at the level of ligands, receptors, or effectors is observed in several types of cancer, including colon, lung, breast, and prostate[7,8]. Wnt proteins bind to transmem-brane Frizzled (FZD) receptors and a variety of co-receptors (LRP4-6, ROR1/2, and RYK)[9] to activate β-catenin-dependent and β-catenin-independent signals. Our understanding of the mechanisms by which Wnt proteins stimulate different signaling responses is incomplete, but they are likely to involve the activation of distinct Wnt receptors in specific cell contexts[8].

A hallmark of β-catenin-dependent Wnt signaling is the stabilization and nuclear translocation of β-catenin, which binds to Tcf/LEF family of transcription factors and exerts effects on the expression of genes that affect cell proliferation and cell fate specification[10]. β-catenin-independent Wnt signals are more diverse, but can be sub-divided into the Planar Cell Polarity (PCP) and the Wnt/Ca²⁺ signaling pathways. PCP signaling involves the small GTPases Rho, which activates Rho-associated kinase, and Rac, which is linked to activation of Jun-N-terminal kinase (JNK) and AP-1 transcription factors and regulates cell migration[10–12]. Wnt/Ca²⁺ signals stimulate Ca²⁺ release from the ER and activate G-proteins, protein kinase C (PKC), and calcium/calmodulin-dependent kinase II, which regulate cancer cell growth, survival, invasion, and angiogenesis[11,13].

Wnt-11 is predominantly a β-catenin-independent Wnt[14] that activates PKC and JNK[15] to increase ATF2-dependent gene expression[16–18] and can also inhibit β-catenin-dependent Wnt signaling[19,20]. Wnt-11 associates with Fzd-7 in Xenopus[21,22], Fzd-5 in zebrafish[23], Fzd-4 in mouse cardiomyocytes[24], and Fzd-4 and Fzd-8 in the developing mouse kidney[24]. The response to Wnt-11 is highly context-dependent and therefore likely also to depend on the presence of Wnt co-receptors[25], among which Wnt-11 has been reported to associate with Ror2 in zebrafish[26] and Ryk in Xenopus[27].

While Wnt-11 is best known for its role during embryonic development[14], it has also been linked to different types of cancer[14,28,29]. In prostate cancer, WNT11 mRNA levels are elevated in a subset of high-grade prostatic tumors, CRPC xenografts, and tumor metastases[28,29]. Inhibition of AR signaling increases WNT11 gene expression, and Wnt-11, in turn, inhibits AR-dependent transcriptional activity and AR-dependent proliferation[28]. Wnt-11 also promotes prostate tumor cell survival, migration, invasion, and neuroendocrine-like differentiation (NED)[29]. However, the receptors that transduce Wnt-11 signals in prostate cancer are not known. Here, we addressed this question, focusing on Wnt-11 receptors required for prostate cancer cell migration and invasion. We find that FZD₈ is a major Wnt-11 receptor in prostate cancer and show that it is upregulated in metastatic disease, where it plays a crucial role in mediating crosstalk between Wnt and TGF-β signaling pathways during the epithelial-to-mesenchymal transition (EMT), which is important for prostate cancer cell migration and invasion.

## Results

**Wnt receptors with increased expression in prostate cancer.** Wnt-11 is elevated in prostate tumors, particularly in patient metastases[29], hormone-depleted LNCaP cells, and castration-resistant tumor xenografts[28]. A variety of proteins bind Wnt ligands, including FZD family members, tyrosine kinase-like receptors, and others[9]. However, it is not known which of them mediate the response to Wnt-11 and play a role in prostate cancer. To identify candidate Wnt-11 receptors, WNT11 and Wnt receptor mRNA expression levels were compared in a panel of prostate cancer cell lines and in hormone-depleted cells. Genes encoding FZD₂₋₅, FZD₈, VANGL1, ROR1, RYK, LGR4, LRP5 and 6, and GPC4 were highly expressed in at least three prostate cancer cell lines (Table 1, Supplementary Fig. 1a and Supplementary Table 6). FZD4 expression was higher in androgen-independent and metastatic prostate cancer cell lines, while FZD2 was upregulated in hormone-depleted LNCaP cells, similar to WNT11. Hormone-depletion also increased expression of FZD7, VANGL2, ROR2, PTK7, and LRP4 mRNA. FZD8 expression was high in metastatic cell lines but reduced in hormone-depleted LNCaP cells (Table 1, Supplementary Fig. 1a, b).

In order to assess the potential relevance of Wnt receptors in patient tumors, in silico analysis was carried out using the Oncomine™ database. FZD4, FZD8, and PTK7, but not FZD2, were more highly expressed in prostate cancer than in benign or normal prostate tissue (Table 2 and Supplementary Fig. 2a). A potential role for these receptors has also been described in other types of cancer[30–33]. Of note, FZD8 mRNA levels are higher in prostate tumor epithelial cells than in luminal epithelial cells from benign prostate[34]. Moreover, FZD8 was upregulated in the highest number of datasets when comparing prostate carcinoma and benign prostate (Supplementary Fig. 2b). Analysis of data from cBioPortal for Cancer Genomics[35] revealed elevated FZD8 (z-score threshold ±1.5) in 10 and 6% of tumors from the

### Table 1 Wnt receptor expression in prostate cancer cell lines

| Receptor | UP in NED | UP in AI PCa | UP in mPCa | Restricted expression |
|---|---|---|---|---|
| **FZD2** | x | x | x | |
| FZD3 | | | | |
| **FZD4** | | x | x | |
| FZD5 | | x | | |
| FZD6 | | | | |
| FZD7 | x | | | |
| **FZD8** | | | x | |
| FZD9 | | x | | |
| FZD10 | | | | x |
| VANGL1 | | x | | |
| VANGL2 | x | | | |
| ROR1 | | | x | |
| ROR2 | x | | | |
| RYK | | x | x | |
| PTK7 | x | | | |
| LGR4 | | | x | |
| LGR5 | | | | x |
| LRP4 | x | | x | |
| LRP5 | x | | | |
| LRP6 | | | | |
| GPC4 | | | x | |
| MuSK | | | | x |

*Note*: Summary of Wnt receptor mRNA expression in prostate cancer (PCa) cell lines with x denoting expression in lines with features of neuroendocrine-like differentiation (NED), androgen-independence (AI), capacity for metastasis (Met) and restricted expression to one line (RE) (for details see Supplementary Fig. 1); FZD2, FZD4 and FZD8 (highlighted in bold) were highly expressed in cells with more aggressive features of PCa; FZD1 was not detected

**Table 2 Wnt receptor expression in prostate cancer cell lines**

| Receptor | UP in PCa | DOWN in PCa |
|---|---|---|
| FZD1 | 1 | 6 |
| FZD2 | 1 | 1 |
| FZD3 | 1 | 1 |
| **FZD4** | **4** | |
| FZD5 | | 1 |
| FZD6 | | |
| FZD7 | 1 | 6 |
| **FZD8** | **6** | |
| FZD9 | | |
| FZD10 | | 2 |
| VANGL1 | | |
| VANGL2 | | 1 |
| ROR1 | | 1 |
| ROR2 | | 6 |
| RYK | | 1 |
| **PTK7** | **2** | |
| LGR4 | | 1 |
| LGR5 | | 2 |
| LRP4 | | 2 |
| LRP5 | | |
| LRP6 | 1 | 1 |
| GPC4 | | 1 |
| MUSK | | 1 |

*Note*: Analysis of Wnt receptor mRNA expression profiles in datasets from Oncomine™; FZD4, FZD8 and PTK7 (highlighted in bold) were upregulated in PCa in more datasets than the other receptors

TGCA[36] and SU2C/PCF Metastatic Prostate Cancer[37] datasets, respectively. Examination of receptor protein levels in the Protein Atlas database (www.proteinatlas.org) further implicated $FZD_8$, revealing low levels in normal prostate and moderate expression in cancer. In contrast, $FZD_4$ and PTK7 levels were similar (low and high, respectively) in prostate and prostate cancer (Supplementary Fig. 2c).

**$FZD_8$ is required for Wnt-11/ATF-2 signaling**. After identifying Wnt receptors with increased expression in prostate cancer, immunofluorescence assays were used to determine which FZD receptors, when transfected, co-localized with Wnt-11. Wnt-11 co-localized strongly with $FZD_8$ and moderately with $FZD_6$ and $FZD_{10}$, but not with other FZD family members (Fig. 1a and Supplementary Fig. 3). To determine which FZD receptors might be capable of transducing Wnt-11 signals, we examined their effects on ATF2-dependent signaling. Wnt-11 activates ATF2 in a variety of contexts, including cardiac tissue morphogenesis[16], Xenopus embryo development[17], and avian facial morphogenesis[18]. Wnt-11/ATF-2 signaling in prostate cancer cells was measured using an ATF2-dependent luciferase reporter[17] (Fig. 1b). ATF2-dependent transcriptional activity in PC-3M cells was significantly enhanced upon transfection of seven members of the FZD family (Fig. 1c). Among these, $FZD_8$ and $FZD_{10}$ showed the strongest induction, which was further increased by Wnt-11 (Fig. 1c). Next, gene reporter assays were carried out in PC-3M cells transfected with siRNAs targeting the more highly expressed FZD family members. FZD8 silencing reduced Wnt-11 activation of ATF2-dependent transcription by 40% and FZD2 silencing reduced it by 20%, whereas silencing FZD3, FZD4, and FZD5 had no effect (Fig. 1d). FZD8 silencing also resulted in a small reduction in FZD5 expression (Supplementary Fig. 4a). However, FZD5 silencing did not affect Wnt-11 activation of ATF2-dependent transcription (Fig. 1d) and an unrelated

Dicer-substrate FZD8 siRNA also inhibited Wnt-11 activation of ATF2-dependent transcription (Supplementary Fig. 4b), consistent with endogenous $FZD_8$, rather than $FZD_5$, mediating the Wnt-11 response. Importantly, the effect of FZD8 siRNA on ATF2 transcriptional activity was rescued by transfection of mouse FZD8 (Supplementary Fig. 4c). Together, these findings indicate that Wnt-11 activation of ATF2-dependent transcription requires endogenous $FZD_8$.

ATF2 is a component of the AP-1 complex and Wnt-11 can also regulate AP-1 signaling[14]. Consistent with this, FZD8 silencing reduced AP-1-dependent gene reporter activity (Fig. 1e). $FZD_8$ can also transduce Wnt/β-catenin signals[32,38]. However, FZD8 silencing did not inhibit β-catenin/Tcf-dependent transcriptional activity in PC-3M cells (Fig. 1f). Moreover, FZD8 silencing reduced expression of ATF2, but not the β-catenin/Tcf target gene AXIN2, which actually increased (Fig. 1g). Together, these data support a role for $FZD_8$ in β-catenin-independent Wnt signaling in prostate cancer.

The association of Wnt proteins with FZD receptors is an essential step in activation of both canonical and non-canonical Wnt signals[39]. To determine if Wnt-11 and $FZD_8$ formed a stable complex, immunoprecipitation (IP) analysis was carried out using PC-3M cells transfected with tagged forms of Wnt-11 and $FZD_8$. The results indicated that Wnt-11 and $FZD_8$ can form a stable complex (Fig. 1h).

**$FZD_8$ is required for cell invasion and EMT**. Wnt-11 is required for prostate cancer cell migration and invasion[29]. To determine the role of $FZD_8$ in this context, cell migration and invasion assays were carried out using PC-3M and another metastatic cell line, DU145. FZD8 silencing reduced cell migration (Fig. 2a and Supplementary Fig. 5a) and invasion (Fig. 2b and Supplementary Fig. 5b) in both cell lines. Of note, FZD8 silencing also had a small effect on cell number, reducing it by up to 20% (Supplementary Fig. 5c, d), but this effect was taken into account when measuring effects on migration and invasion. Stable FZD8 knockdown using two different lentiviral shFZD8 constructs also reduced prostate cancer cell invasion (Supplementary Fig. 5e). The extent of FZD8 silencing was confirmed by q-RT-PCR (Supplementary Fig. 5f) and effects on cell number (Supplementary Fig. 5g) were taken into account. Together, these results indicate that $FZD_8$ contributes to the migratory and invasive activities of metastatic prostate cancer cells. A recent study reported small molecule inhibitors of Wnt signaling that target the WNT-binding site on $FZD_8$[40]. Two of these inhibitors reduced PC-3M cell migration (Fig. 2c) without affecting cell proliferation (Supplementary Fig. 5h).

The EMT is a reversible process in which epithelial cells acquire mesenchymal properties by altering their morphology, cellular architecture, adhesion, and migratory capacities[41]. Since Wnt signals play roles both in EMT and tumor progression[41], we hypothesized that Wnt11- $FZD_8$ signals drive EMT. To test this, the effect of FZD8 silencing on EMT-associated gene expression was examined (Fig. 2d and Supplementary Fig. 6a). FZD8 silencing in PC-3M cells reduced expression of the mesenchymal genes CDH2 and VIM, and the mesenchymal transcription factors SNAI1, TWIST1, and ZEB1, but not SNAI2. In addition, FZD8 silencing increased expression of the epithelial cell marker CLDN1 (Fig. 2d). FZD8 silencing had similar effects on EMT-related genes in DU145 cells (Supplementary Fig. 6a). However, it did not affect CLDN1 or TWIST1 and reduced SNAI2 (Supplementary Fig. 6a), possibly reflecting the more epithelial nature of DU145 cells or the different extent of FZD8 silencing achieved (Fig. 2f and Supplementary Fig. 6c). Western blot analysis indicated that FZD8 silencing also affected EMT at the

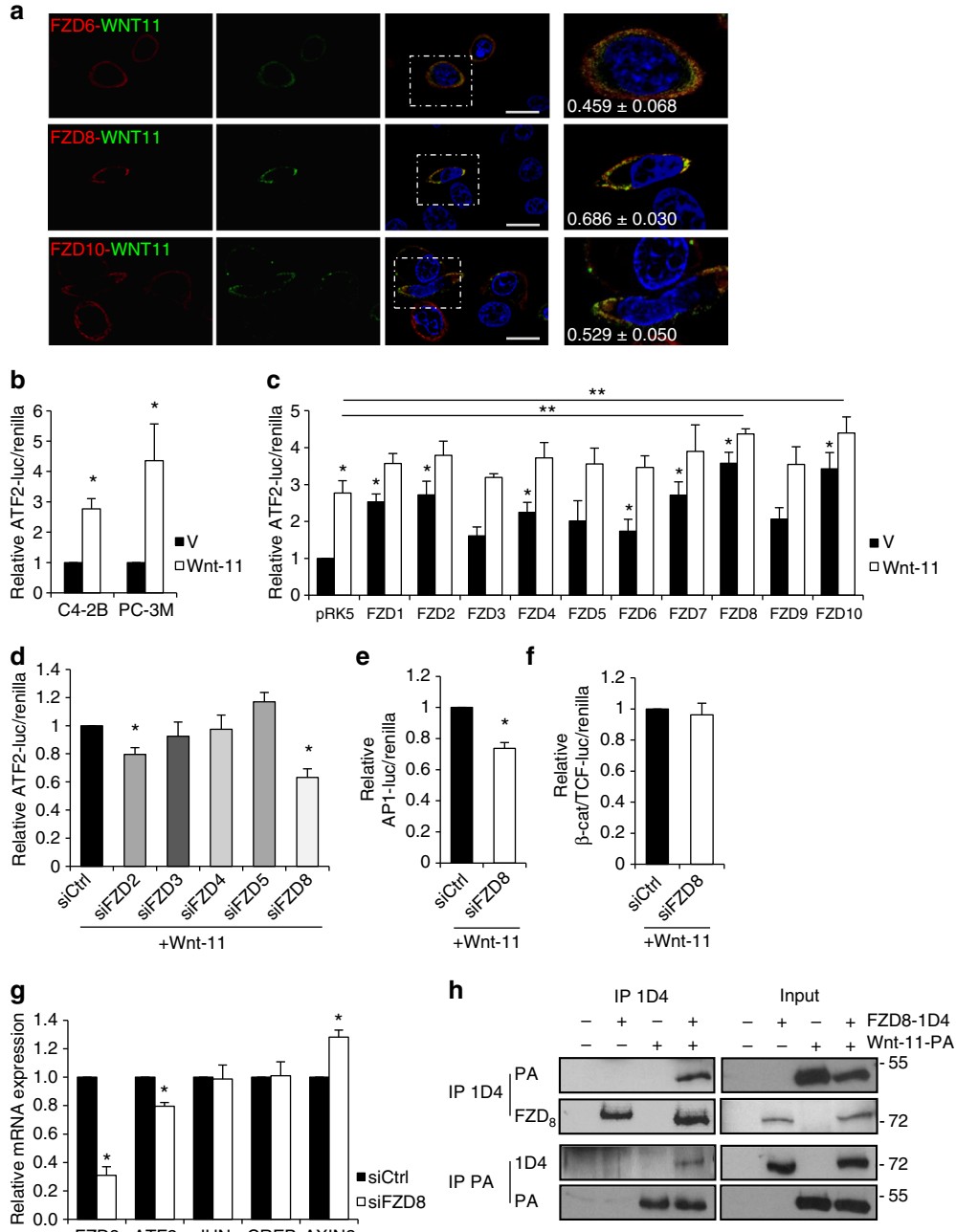

**Fig. 1** FZD8 is required for Wnt-11/ATF-2 signaling in prostate cancer. **a** Confocal microscopy analysis of PC-3M cells transfected with 1D4-tagged $FZD_6$, $FZD_8$, or $FZD_{10}$ (red) and Wnt-11 (green) for 24 h (for other FZD family members, see Supplementary Fig. 2); low magnification images of single color and dual channels and high magnification images of single cells are shown; anti-1D4 and goat anti-Wnt-11 (R&D) were used to detect FZDs and Wnt-11, respectively; blue staining shows cell nuclei (DAPI), images are representative of three independent experiments, scale bars 25 μm. Quantification of colocalization was determined by ImageJ (see Methods) using 10 cells per experiment. Numbers in the images correspond to average Pearson correlation coefficient -/+ standard deviation. **b** Relative ATF2 luciferase/renilla activity in C4-2B and PC-3M cells transfected with empty vector pcDNA (V) or Wnt-11; results are normalized to empty vector. **c** Relative ATF2 luciferase/renilla activity in C4-2B cells transfected with empty vector pRK5 (V) or Wnt-11 and FZD 1-10; results are normalized to empty vector. **d** Relative ATF2 luciferase/renilla activity in PC-3M cells transfected with control siRNA (siCtrl) or the indicated siRNAs and then with Wnt-11; results are normalized to siCtrl. **e** Relative AP1 luciferase/renilla activity in PC-3M cells transfected with siCtrl or siFZD8 and then with Wnt-11; results are normalized to siCtrl. **f** Relative β-catenin/TCF activity (TOPFlash/FOPFlash) in PC-3M cells transfected with siCtrl or siFZD8 and then with Wnt-11; results are normalized to siCtrl. **g** Q-PCR analysis showing mRNA expression of the indicated genes, relative to 36B4, in PC-3M cells transfected with control (siCtrl) and FZD8 siRNAs. **h** Western blots of anti-1D4 and anti-PA immunoprecipitates (IP) and extracts (Input) from PC-3M cells transfected with 1D4-tagged $FZD_8$ and PA-tagged Wnt-11 plasmids. Extracts were probed with PA, 1D4, or $FZD_8$ (LS bio) antibodies; blots are representative of three independent experiments. Error bars in **b**–**g** show SD from four independent experiments (*$p < 0.05$, **$p < 0.001$ by Student's $t$-test or ANOVA with Tukey post hoc test where required)

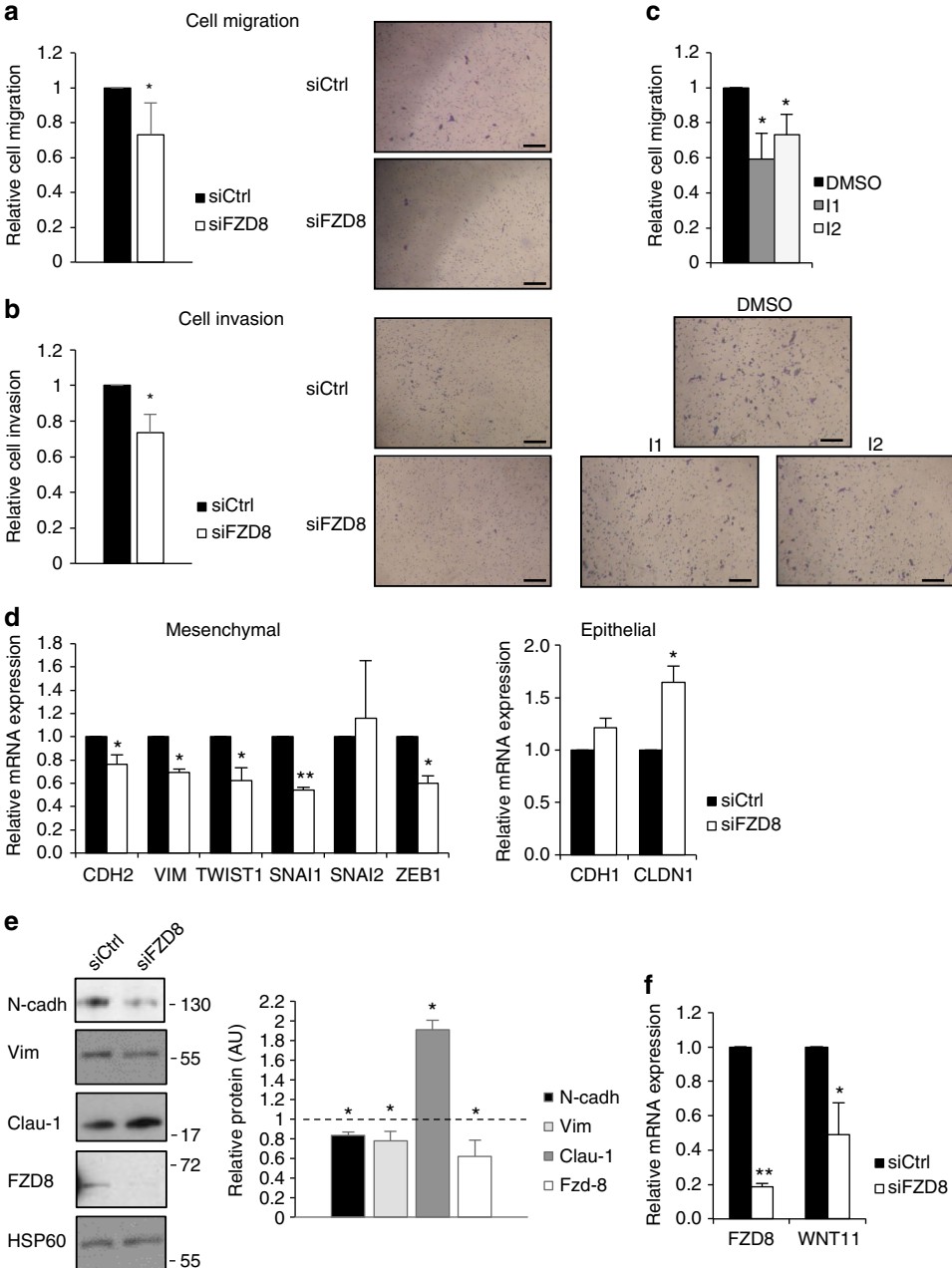

**Fig. 2** FZD8 is required for prostate cancer cell migration, invasion, and expression of epithelial–mesenchymal transition (EMT) genes. **a** Migration assays for PC-3M cells transfected with control (siCtrl) and FZD8 siRNAs; values are relative to siCtrl and normalized to viable cell number of transfected cells plated in parallel (Supplementary Fig. 5c); error bars show SD for six independent experiments (*$p < 0.05$ by Student's $t$-test). Representative images are on the right, scale bar 100 μm. **b** Invasion assays for PC-3M cells transfected with control (siCtrl) and FZD8 siRNAs, values are relative to siCtrl and normalized to viable cell number (Supplementary Fig. 5d); error bars show SD for four independent experiments (*$p < 0.05$ by Student's $t$-test). Representative images are on the right, scale bar 100 μm. **c** Migration assays for PC-3M cells treated with DMSO or inhibitors 1 (I1) and 2 (I2) at 10 μM for 24 h; values are relative to DMSO and normalized to viable cell number (Supplementary Fig. 5h), error bars show SD for four independent experiments (*$p < 0.05$ by ANOVA with Tukey post hoc test). Representative images are below the graph, scale bar 100 μm. **d** Q-PCR analysis showing expression levels of the indicated EMT genes, normalized to 36B4, in PC-3M cells transfected with control (siCtrl) or FZD8 siRNAs; error bars show SD for four independent experiments (*$p < 0.05$, **$p < 0.001$ by Student's $t$-test). **e** Extracts from PC-3M cells transfected with control or FZD8 siRNAs were blotted for the indicated proteins; graph shows average relative protein levels, as determined by densitometry analysis of blots from three independent experiments (*$p < 0.05$ by Student's $t$-test), normalized to HSP60 and relative to control siRNA (siCtrl). **f** Relative expression levels of FDZ8 and WNT11 measured by Q-PCR in PC-3M cells transfected with control (siCtrl) and FZD8 siRNAs; error bars represent SD of four independent experiments (*$p < 0.05$, **$p < 0.001$ by Student's $t$-test)

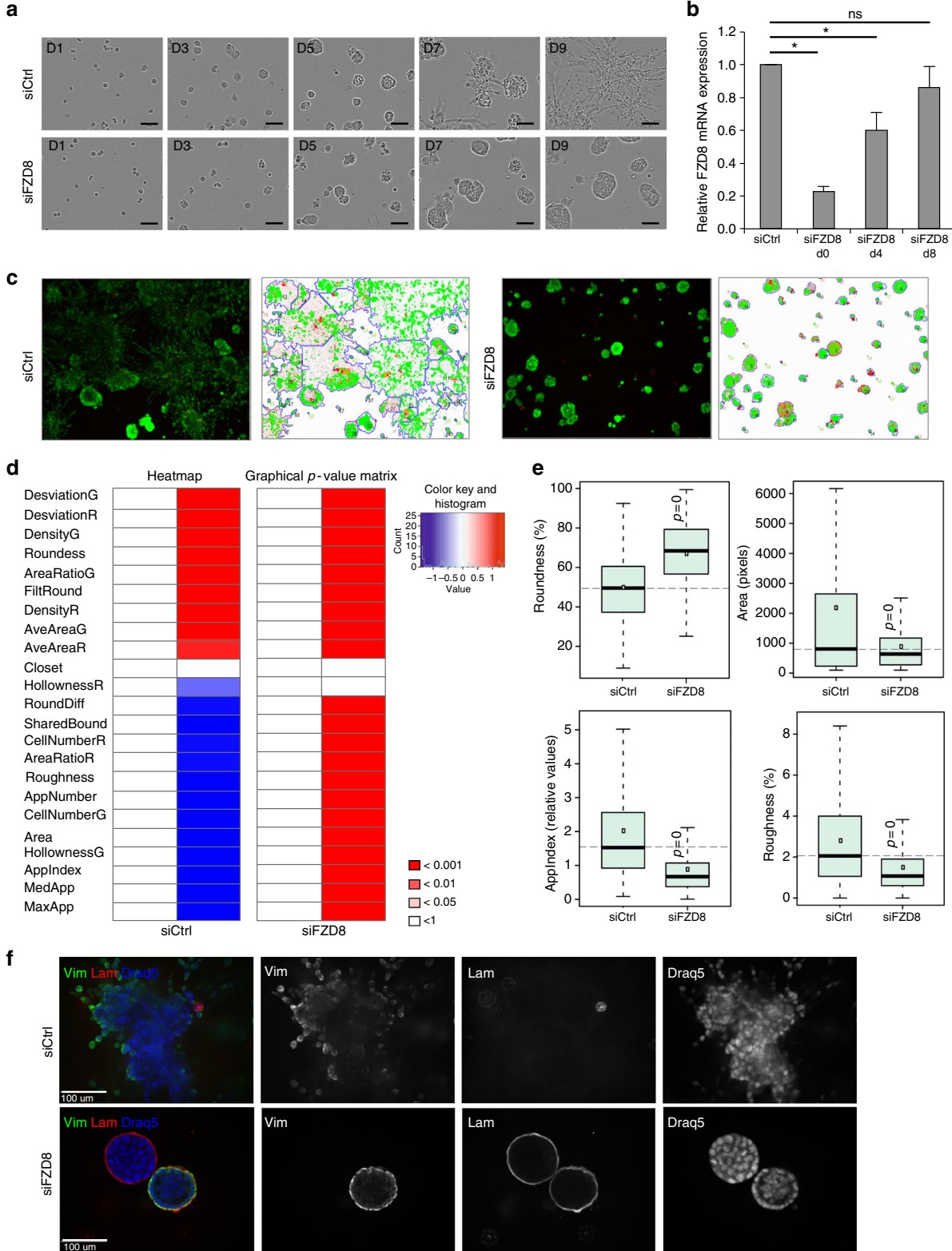

**Fig. 3** FZD8 is required for prostate cancer cell invasion in organotypic 3D cultures. **a** Representative images from live-cell imaging of 3D cultures of control (siCtrl) and FZD8-silenced PC-3M cells at days 1–9; scale bar 100 μm. **b** FZD8 expression levels in siRNA-transfected cells at days 0, 4, and 8 of 3D culture. Error bars indicate SD from three independent experiments (*$p < 0.05$, **$p < 0.001$ by ANOVA with Tukey post hoc test, ns indicate non-significant). **c** Representative segmentation of live-cell spinning disk confocal images of FZD8-silenced PC-3M cell organoids cultured in 3D for 9 days. Organoids were segmented and analyzed by AMIDA; apoptotic cells are in red (ethidium homodimer-1) and live cells in green (calcein). **d** Heatmaps and graphical *p*-value matrix of morphometric parameters measured by AMIDA and found to be altered by FZD8 silencing (red: increased, and blue: decreased). *p*-values displayed in the figure are Bonferroni-corrected from *t*-tests, comparing siFZD8 and siCtrl. **e** Box and whisker plots of selected parameters from heatmaps; $p = 0$ indicates $p < 0.001$. For explanation of the morphometric parameters, see Supplementary Table 4. **f** Confocal microscopy analysis of organoids derived from siCtrl and FZD8-silenced PC-3M cells at day 9 of growth in 3D culture; immunostaining for vimentin (Vim) is shown in green and for laminin-α1 (Lam) in red, blue staining shows cell nuclei (Draq5), scale bar 100 μm

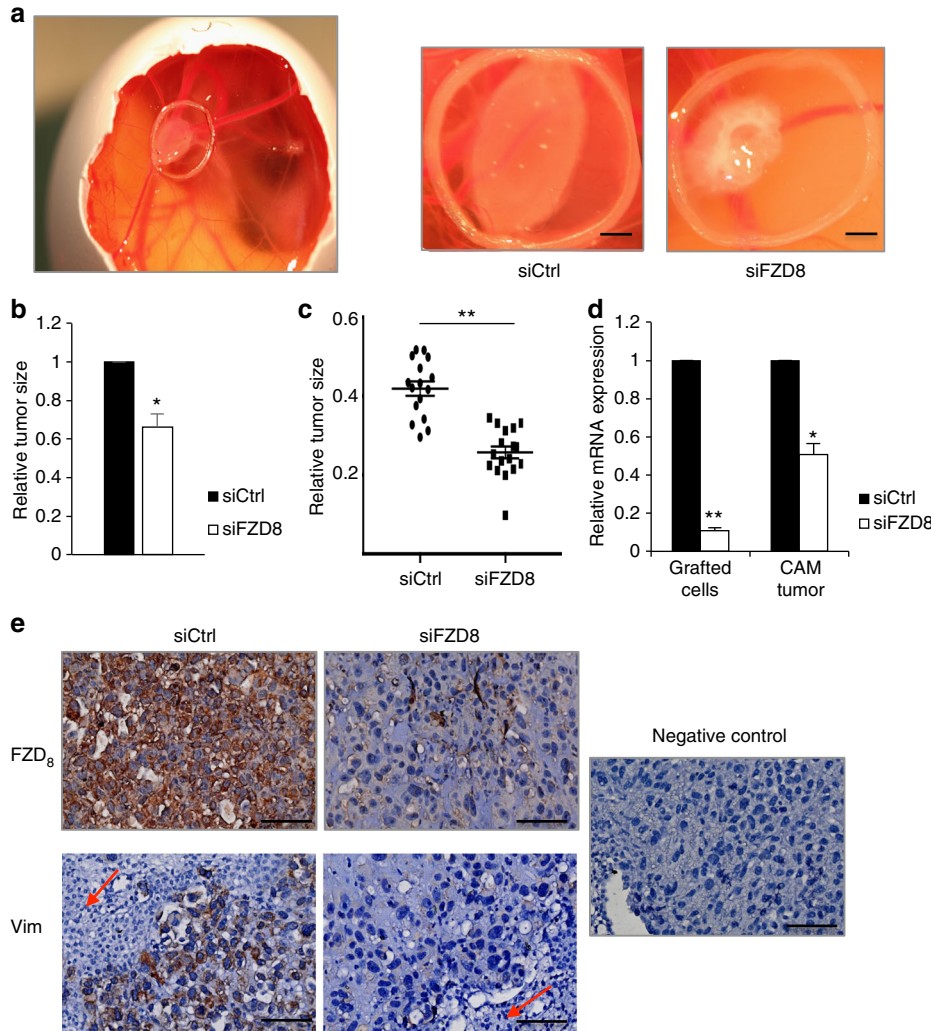

**Fig. 4** FZD8 is required for tumor growth in vivo. **a** Representative pictures of PC-3-derived tumors implanted on the CAM for 3 days. Left, overview image of the egg; right, representative tumors from PC-3 cells transfected with control (siCtrl) and FZD8 siRNAs. Implantation area is confined by a plastic ring; scale bars ~5 mm. **b** Average tumor size, as determined by ImageJ, relative to control tumors. Error bars indicate SD from three independent experiments, each with 12–15 eggs per condition (*$p < 0.05$ by Student's $t$-test). **c** Boxplot of tumor size in a representative experiment with 12–15 eggs per condition (**$p < 0.001$ by Student's $t$-test). **d** Relative FZD8 mRNA levels were determined by q-RT-PCR at day 0, when cells were implanted on the CAM and at the endpoint, when CAM tumors were excised from the egg. Error bars indicate SD of three independent experiments (*$p < 0.05$, **$p < 0.001$ by Student's $t$-test). **e** Immunohistochemical staining of CAM tumors. Representative images of sections immunostained for $FZD_8$ and vimentin in control (siCtrl) and FZD8-silenced (siFZD8) CAM tumors; arrows show chick embryo cells, which can be distinguished from PC-3 cells by the absence of vimentin staining, scale bar 50 μm

protein level, reducing the levels of N-cadherin and vimentin and increasing that of claudin 1 (Fig. 2e). Furthermore, bioinformatics analysis indicated that both FZD8 and WNT11 expression correlated with SNAI1, SNAI3, TWIST1, and TWIST2, and negatively correlated with CDH1 and CTNNB1 (Supplementary Fig. 6b). WNT11 has itself been described as an EMT gene in kidney epithelial cells[42], and we observed that FZD8 silencing also reduced WNT11 expression (Fig. 2f and Supplementary Fig. 6c), and found a positive correlation between FZD8 and WNT11 expression in the MSKCC dataset (Supplementary Fig. 6d). These results suggest a positive-feedback loop in which $FZD_8$ signals lead to increased WNT11 expression.

**FZD8 is required for cell invasion in 3D organotypic cultures.** To investigate the role of FZD8 in prostate cancer cell invasion further, we used an organotypic cell culture model in which acinar structures (organoids) are formed that display physiologically relevant cell–cell and cell–matrix interactions, epithelial polarization and differentiation, recapitulating human cancer histology[43]. PC-3 and PC-3M metastatic prostate cancer cells initially differentiate into hollow organoids (days 4–5) and later spontaneously de-differentiate into invasive stellate structures (days 8–12)[43]. Cells were transfected with FZD8 siRNA and cultured as organoids, monitoring morphology, polarization, and growth for up to 9 days using the IncuCyte® system. Cells transfected with control siRNA initially matured into well-differentiated organoids and then formed invasive and multi-cellular structures at days 7–8 (Fig. 3a and Supplementary Fig. 7a). In contrast, FZD8-silenced cells matured into well-differentiated organoids but did not form invasive structures (Fig. 3a and Supplementary Fig. 7a). Of note, these effects were completely evident at the endpoint of the assay, despite recovery of basal FZD8 mRNA expression by day 8 (Fig. 3b and Supplementary Fig. 7b).

To quantify the phenotypic changes, at least 1000 organoids were examined using high-content, automated morphometric

image data analysis (AMIDA) software, which allows segmentation and quantitative measurement of images with different shapes, sizes, and textures. Organoids were live-stained with calcein and ethidium homodimer at the endpoint of the 3D culture, visualized by confocal microscopy and images segmented (Fig. 3c and Supplementary Fig. 7c), and analyzed using AMIDA. FZD8 silencing significantly reduced the severity (AppIndex) and length (MaxApp) of invasive multicellular structures, accompanied by reductions in the numbers of small filopodia-like cellular extensions (Roughness) and organoid size (Area), as well as by a rounder shape (Roundness) (Fig. 3d, e and Supplementary Fig. 7d, e). Organoid morphology was also examined by

immunostaining. Laminin-α1 was not detectable in control organoids but was clearly observed in FZD8-silenced organoids, indicating the presence of a basal lamina characteristic of well-differentiated structures (Fig. 3f and Supplementary Fig. 7f). On the other hand, vimentin staining highlighted invasive cells migrating from control cell organoids but not from FZD8-silenced organoids, reflecting a reduction in invasive properties of the latter (Fig. 3f and Supplementary Fig. 7f). Interestingly, some FZD8-silenced organoids showed a total absence of vimentin staining (Fig. 3f). These 3D cell culture data further support a role for FZD8 in promoting tumor cell invasion in prostate cancer.

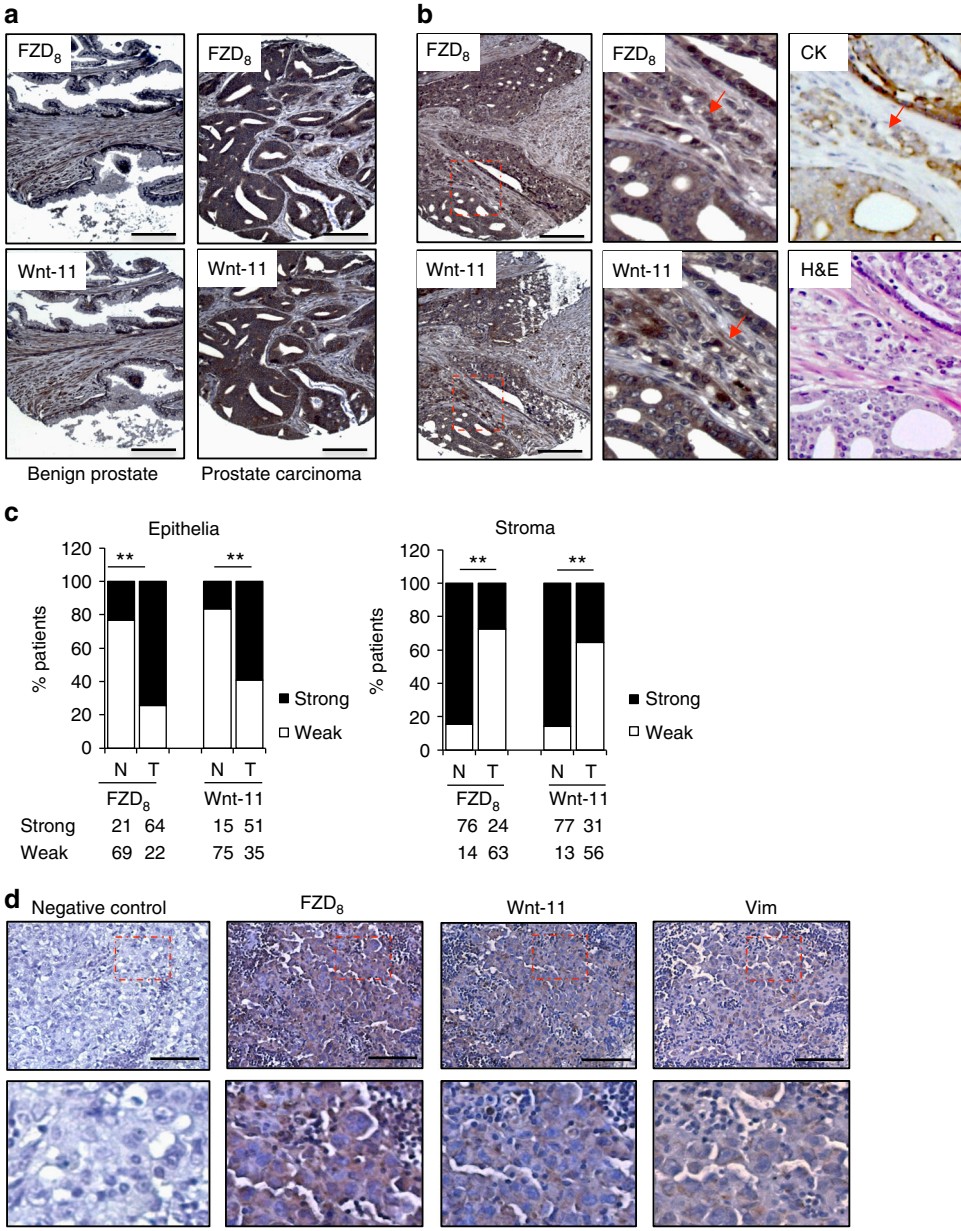

**Fig. 5** $FZD_8$ and Wnt-11 expression correlates with prostate cancer progression. **a** Immunohistochemical staining of $FZD_8$ and Wnt-11 in adjacent sections of prostate cancer (Gleason $4 + 3$) and an area of benign prostate from the same patient; scale bar 25 μm. **b** Immunohistochemical staining of $FZD_8$ and Wnt-11 in a section of prostate cancer (Gleason $4 + 3$) from another patient; scale bar 25 μm. High magnification images for $FZD_8$, Wnt-11, epithelial pan-cytokeratin (CK), and H&E are also shown. Arrows show disseminated tumor cells positive for $FZD_8$, Wnt-11, and epithelial cytokeratins. **c** Stratification of $FZD_8$ and Wnt-11 expression in cancer (T) and benign (N) epithelia and stroma (\*\*$p < 0.001$ by Pearson Chi-square test with correction). **d** Immunohistochemical staining for $FZD_8$, Wnt-11, and vimentin in sections of a PC-3M cell lymph node metastasis from a mouse orthotopic xenograft; scale bars 50 μm. Squares highlight regions shown at higher magnification below

| Table 3 Correlation analysis of $FZD_8$ and Wnt-11 expression | | | |
|---|---|---|---|
| | $FZD_8$ | Wnt-11 | Chi-square; Fisher exact |
| **Cancer** | | | |
| | | Low | High | <0.0001; 0.000071 |
| | Low | 17 | 5 | |
| | High | 18 | 46 | |
| **Benign** | | | |
| | | Low | High | No value;0.039 |
| | Low | 61 | 8 | |
| | High | 14 | 7 | |
| **Cancer stroma** | | | |
| | | Low | High | 0.0063; 0.011 |
| | Low | 46 | 17 | |
| | High | 10 | 14 | |
| **Benign stroma** | | | |
| | | Low | High | No value;0.027 |
| | Low | 5 | 9 | |
| | High | 8 | 68 | |

Note: $p < 0.001$ by Pearson Chi-square test with correction and Fisher's exact test, two-sided

**FZD8 silencing reduces tumor growth in vivo**. To validate the functional role of FZD8 in prostate cancer cells in vivo, we used the chorioallantoic membrane (CAM) model. The CAM is a highly vascular membrane in chicken eggs that enables efficient tumor cell grafting and growth, mimicking a physiological cancer cell environment[44]. Control and FZD8-silenced PC-3 cells were grafted onto the exposed CAM at developmental day 7 (EDD7). At EDD10 the tumors formed by FZD8-silenced PC-3 cells were significantly smaller than those formed by control cells (Fig. 4a–c). Evaluation of FZD8 levels at the moment of cell grafting and when tumors were excised indicated that silencing was maintained during the course of the assay (Fig. 4d). Analysis of the expression of vimentin, which contributes to prostate cancer invasiveness[45,46], indicated that tumors in which $FZD_8$ was silenced exhibited lower levels of vimentin than control tumors, consistent with the role of FZD8 in prostate tumor cell invasion (Fig. 4e). Of note, immunostaining for vimentin distinguishes prostate cancer cells from CAM cells, which are negative for this marker (Fig. 4e).

**$FZD_8$ and Wnt-11 correlate with prostate cancer progression**. In order to evaluate FZD8 and WNT11 gene expression during prostate cancer progression, bioinformatics analyses were performed using the MSKCC dataset[47]. These revealed a clear upregulation of FZD8 in prostate cancer, compared to normal prostate, and in prostate cancer metastases, compared to primary tumors. Both FZD8 and WNT11 were upregulated in high Gleason score tumors and in tumors that had spread to lymph nodes (Supplementary Fig. 8a). Expression of WNT11 correlated significantly ($p = 0.014$) and FZD8 showed a trend ($p = 0.057$) with increased biochemical recurrence (Supplementary Fig. 8b). Further analysis indicated that elevated FZD8 and WNT11 expression was more prevalent in patients with recurrent disease, as compared to disease-free patients ($p = 0.049$, Fisher's exact test, two-sided) (Supplementary Fig. 8c), supporting the relevance of WNT-11-FZD8 signaling to prostate cancer progression.

To confirm the in silico data, immunohistochemistry for $FZD_8$ and Wnt-11 was carried out in tissue arrays (TMAs) comprising sections of benign and malignant prostate from prostate cancer patients (Supplementary Table 5). This revealed significantly higher levels of $FZD_8$ and Wnt-11 in tumor cells, compared to in benign epithelium (Fig. 5a–c, Supplementary Table 7) and correlations in the levels of $FZD_8$ and Wnt-11 in benign and tumor epithelia and stroma (Table 3). $FZD_8$ and Wnt-11 were

also both significantly lower in tumor stroma than in the benign stroma (Fig. 5c, Supplementary Table 7), suggesting that elevated expression in cancer epithelial cells is accompanied by reduced expression in tumor-associated stroma. Even though $FZD_8$ and Wnt-11 levels were higher in prostate cancer than in benign prostatic epithelium, there was no significant correlation with Gleason score in this patient cohort (Supplementary Table 8). $FZD_8$ and Wnt-11 were also detected in lymph node metastases after orthotopic implantation of mice with PC-3M cells (Fig. 5d), consistent with a role for Wnt-11/$FZD_8$ signaling in metastasis. Vimentin staining was used to distinguish prostate cancer cells from host cells.

**$FZD_8$ regulates TGF-β/Smad signaling**. Crosstalk between TGF-β and Wnt signaling during development has been studied extensively[48,49]. The best-defined venue for crosstalk is the nucleus, where Smad proteins have been reported to associate with β-catenin/Tcf/LEF complexes to regulate gene expression[48]. Smads also associate with AP-1 transcription factors, which mediate β-catenin-independent Wnt signals[50]. TGF-β is a recognized master regulator of EMT[51], regulating expression of SNAIL/TWIST/ZEB, which control cadherin switching, matrix metalloproteinases (MMPs), plasminogen activator inhibitor-1 (PAI1), and vimentin. Since $FZD_8$ is required for Wnt-11/ATF2 signaling and EMT, we hypothesized that it plays a role in TGF-β signaling. Consistent with this, FZD8 silencing reduced TGF-β activation of a Smad-dependent gene reporter (Fig. 6a and Supplementary Fig. 9a). Inhibition of TGF-β/Smad-dependent transcription was also observed using an unrelated Dicer-substrate FZD8 siRNA (Supplementary Fig. 9b). Although FZD8 silencing reduced TGF-β activation of the Smad-dependent gene reporter, it did not block it completely, suggesting FZD8 is not the only effector of TGF-β in this context. FZD8 silencing also reduced expression of MMP9 and PAI1 (Fig. 6b and Supplementary Fig. 9c) and TGF-β-dependent increases in SMAD2 phosphorylation and SMAD2/3 levels (Fig. 6c and Supplementary Fig. 9d).

Since TGF-β signaling increases cell invasion, we determined the effect of FZD8 silencing on TGF-β-induced cell invasion. FZD8 silencing significantly reduced invasion both in PC-3M and DU145 cells (Fig. 6d and Supplementary Fig. 9e) without affecting cell number (Supplementary Fig. 9f, g). To determine if the effects of FZD8 silencing on EMT gene expression were mediated via inhibition of TGF-β signaling, we examined the expression of EMT genes upon TGF-β treatment. In DU145 cells, TGF-β increased VIM, SNAI1/2, and ZEB1 and reduced CDH1 (Supplementary Fig. 10a). TGF-β also increased FZD8 and WNT11 expression (Supplementary Fig. 10c). FZD8 silencing inhibited TGF-β-induced expression of VIM, SNAI1/2, ZEB1, MMP9, and PAI-1 and increased expression of CLDN1 (Fig. 6e). Together, these results indicate that FZD8 plays a role in TGF-β−mediated cell invasion and EMT gene expression.

**Association of $FZD_8$ with TGF-β receptors**. Since $FZD_8$ is involved both in Wnt-11/ATF-2 and TGF-β/Smad signaling and ATF2 has been reported to associate with Smad3[52], we examined the possibility that Wnt-11/$FZD_8$ signaling activates TGF-β/Smad signaling via ATF2. To assess the contribution of ATF2, we used Δ-ATF2, a dominant-negative form of ATF2 that binds and inhibits ATF2 and its AP-1 family partners[53]. Δ-ATF2 significantly reduced TGF-β activation of Smad-dependent transcription (Supplementary Fig. 11a), indicating that ATF2 and/or other AP-1 family members are required for TGF-β/Smad signaling. Inhibition was also observed using a reporter containing Smad and AP-1-binding sites (3TP-lux, Supplementary Fig. 11b).

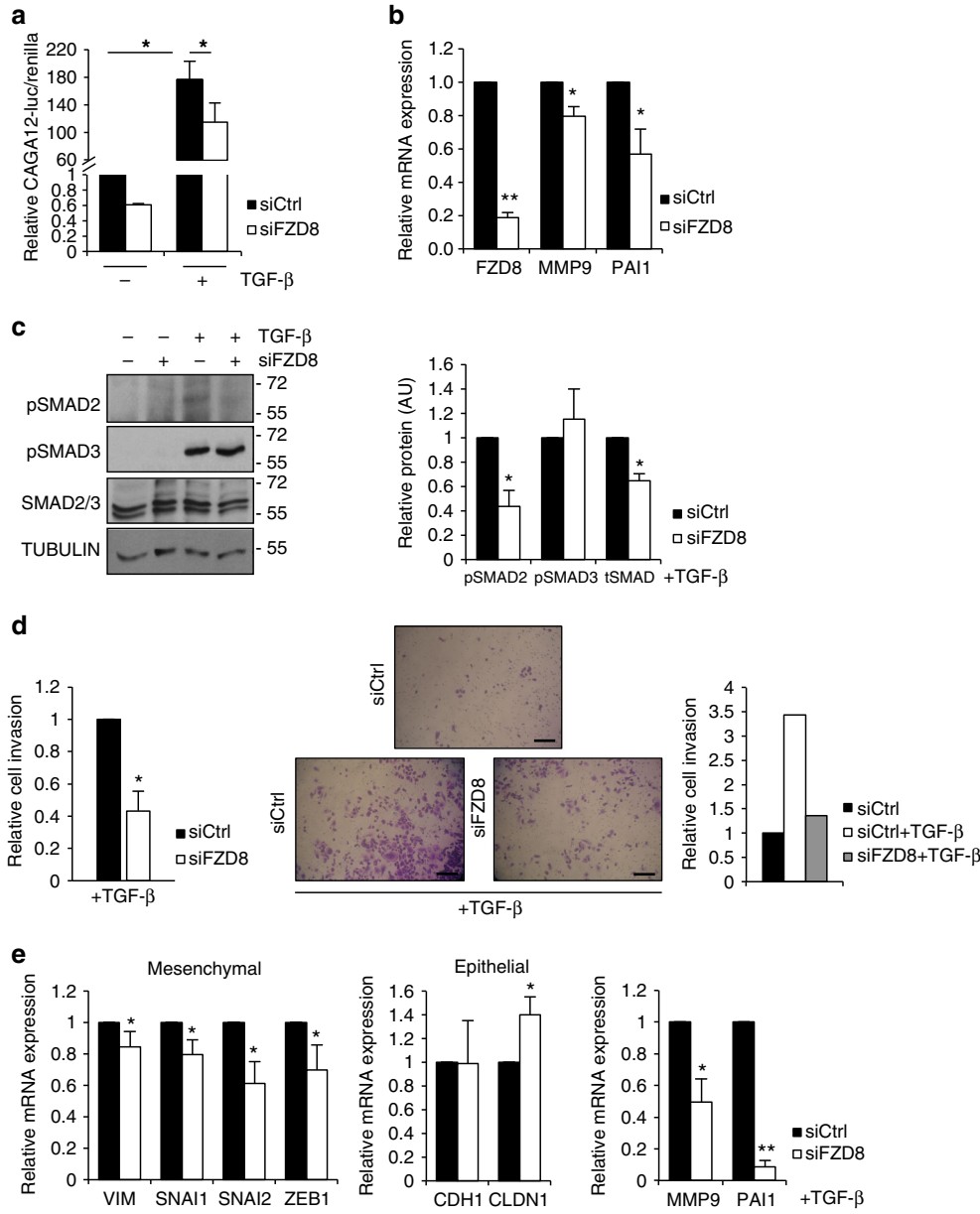

**Fig. 6** FZD8 regulates TGF-β/Smad signaling. **a** Relative CAGA12-luciferase/renilla activity in PC-3M cells transfected with control (siCtrl) or FZD8 siRNAs and treated $-/+1$ ng ml$^{-1}$ TGF-β for 24 h; error bars show SD from four independent experiments (*$p < 0.05$ by ANOVA with Tukey post hoc test). **b** Q-PCR analysis showing expression levels of the indicated genes, normalized to 36B4, in PC-3M cells transfected with control (siCtrl) or FZD8 siRNAs. Error bars show SD of four independent experiments (*$p < 0.05$, **$p < 0.001$ by Student's $t$-test). **c** Western blots of extracts from PC-3M cells transfected with control or FZD8 siRNAs and treated with 1 ng ml$^{-1}$ of TGF-β for 30 min were probed for pSMAD2, pSMAD3, and SMAD2/3 and β-tubulin as a loading control; graph on the right shows relative levels of expression, as determined by densitometry analysis of bands from TGF-β-treated cell extracts, normalized to β-tubulin, and relative to control siRNA. Error bars indicate SD from three independent experiments (*$p < 0.05$ by Student's $t$-test). **d** Invasion assays of control and FZD8-silenced PC-3M cells treated with 5 ng ml$^{-1}$ of TGF-β for 48 h. Left: relative cell invasion normalized to siCtrl in the presence of TGF-β, taking into account effects of silencing on cell number (Supplementary Fig. 9f); error bars represent SD of three independent experiments (*$p < 0.05$ by Student's $t$-test). Middle: representative pictures of invaded cells, scale bar 100 μm. Right: representative experiment. **e** Q-PCR analysis showing mRNA expression of the indicated genes, relative to 36B4, in DU145 cells transfected with control (siCtrl) and FZD8 siRNAs and treated with 1 ng ml$^{-1}$ TGF-β for 24 h; error bars show SD for four independent experiments (*$p < 0.05$, **$p < 0.001$ by Student's $t$-test)

Moreover, Δ-ATF2 reduced TGF-β-induced expression of VIM, SNAI2, and ZEB1 and increased expression of CDH1 in DU145 cells (Supplementary Fig. 11c), suggesting ATF2 is required for expression of a subset of TGF-β-regulated EMT genes. Thus, FZD$_8$ may promote TGF-β/Smad-dependent signaling via activation of ATF2. Given the reported association of ATF2 with Smad3[52], we wished to determine if this was affected by

FZD8 silencing. However, we were unable to detect a stable complex between ATF2 and Smad3 by IP (Supplementary Fig. 11d).

We next hypothesized that crosstalk between FZD$_8$ and TGF-β signaling might take place at the membrane. TGF-β signaling is initiated by ligand binding to two transmembrane receptor kinases (TGFβRI and RII), with ligand binding to TGFβRII a

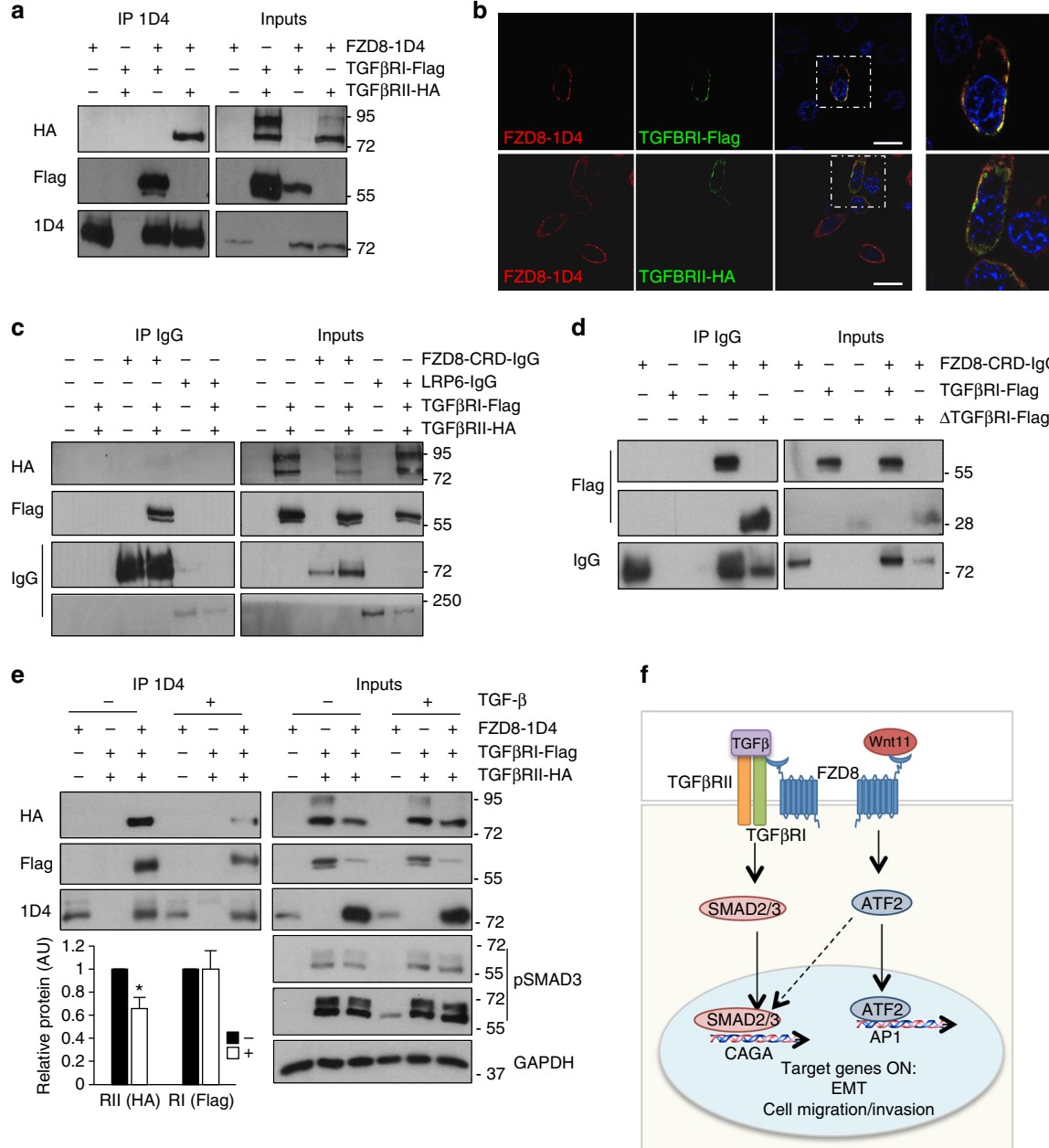

**Fig. 7** FZD8 associates with TGF-β receptors. **a** Western blots of anti-1D4 immunoprecipitates (IP) and extracts (inputs) from PC-3M cells transfected for 24 h with 1D4-tagged FZD$_8$, Flag-tagged TGFβRI, and HA-tagged TGFβRII plasmids were probed for TGFβRII (HA), TGFβRI (Flag), and FZD$_8$ (1D4); blots are representative of three independent experiments. **b** Confocal immunofluorescence analysis of PC-3M cells transfected with 1D4-tagged FZD$_8$ (red) and Flag-tagged TGFβRI or HA-tagged TGFβRII (green) for 24 h. Images are representative of three independent experiments; 1D4 epitope or goat anti-FZD$_8$ (Aviva) were used to detect FZD$_8$ and Flag and HA epitope tag antibodies were used to detect TGFβRI and TGFβRII, respectively; blue staining shows cell nuclei (DAPI); scale bar 25 μm. **c** Western blots of Protein A/G-agarose pull-downs (IP IgG) and extracts (inputs) from PC-3M cells transfected for 24 h with FZD$_8$-CRD-IgG, LRP6-IgG, Flag-tagged TGFβRI, and HA-tagged TGFβRII plasmids were probed for TGFβRII (HA), TGFβRI (Flag), and LRP6 or FZD$_8$ (IgG); blots are representative of three independent experiments. **d** Western blots of Protein A/G-agarose pull-downs (IP IgG) and extracts (inputs) from PC-3M cells transfected for 24 h with FZD$_8$-CRD-IgG, Flag-tagged TGFβRI, and Flag-tagged TGFβRI extracellular domain (ΔTGFβRI) plasmids were probed for TGFβRI (Flag) and FZD$_8$ (IgG); blots are representative of three independent experiments. **e** Western blots of anti-1D4 immunoprecipitates (IP) and extracts (inputs) from PC-3M cells transfected for 24 h with plasmids encoding 1D4-tagged FZD$_8$, Flag-tagged TGFβRI, and HA-tagged TGFβRII, treated −/+1 ng ml$^{-1}$ TGF-β for 30 min were probed for TGFβRII (HA), TGFβRI (Flag), and FZD$_8$ (1D4); extracts were also probed for pSMAD3 and GAPDH as a loading control. Blots are representative of three independent experiments; graph shows average TGFβRII (HA) and TGFβRI (Flag) levels in 1D4 IPs, as determined by densitometry, in control (−) and TGF-β-treated (+) cells transfected with all three receptors, normalized to 1D4 from three independent experiments (*$p < 0.05$ by Student's $t$-test). **f** Cartoon depicting crosstalk of Wnt-11 and TGF-β signaling at the level of the receptors and transcription factors. Dashed arrow indicates potential regulation of SMAD2/3 by ATF-2, based on other published studies (see text)

prerequisite for binding to TGFβRI[54]. To test if FZD$_8$ associated with TGFβ receptors, PC-3M cells were transfected with plasmids expressing epitope-tagged FZD$_8$, TGFβRI, and RII and subjected to IP analysis. A stable interaction was observed between FZD$_8$ and both TGFβRI and TGFβRII (Fig. 7a). In addition, FZD$_8$ partially co-localized with both TGFβRI and TGFβRII in cells (Fig. 7b). FZD$_8$ has an extracellular N-terminal cysteine-rich domain (CRD) that binds to Wnt proteins. To determine if the FZD$_8$ CRD was involved in the association of FZD$_8$ with TGFβ receptors, we used a plasmid encoding FZD$_8$ CRD fused to IgG. This fusion protein associated with TGFβRI but not with TGFβRII (Fig. 7c), suggesting FZD$_8$ associates with TGFβRI and its interaction with TGFβRII (Fig. 7a) is mediated by endogenous TGFβRI. The extracellular domain of LRP6 fused to IgG did not bind either TGFβRI or TGFβRII (Fig. 7c). A tagged form of the TGFβRI extracellular domain readily associated with FZD$_8$ CRD-IgG (Fig. 7d), consistent with the interaction between FZD$_8$ and TGFβRI involving the extracellular domains of both proteins. Together, these results indicate that the association between FZD$_8$ and the TGF-β receptor complex involves the FZD$_8$ CRD and the extracellular domain of TGFβR1. Next, experiments were performed in cells treated with or without exogenous TGF-β. Western blotting for phosphorylated SMAD3 indicated that transfection of TGFBR1/2 was sufficient to activate TGF-β signaling and that this was not affected by exogenous TGF-β (Fig. 7e). Treatment with TGF-β reduced the interaction between FZD$_8$ and TGFβRII, whereas there was no significant effect on its association with TGFβRI (Fig. 7e). These results are consistent with FZD$_8$ interacting with TGFβRII via TGFβRI and with TGF-β treatment reducing binding of TGFβRII to the FZD$_8$–TGFBR1 complex. Taken together, these observations support a model in which FZD$_8$, by interacting both with Wnt-11 and TGFβRI, is able to play a pivotal role in integrating Wnt and TGF-β signals to drive EMT and invasion in prostate cancer (Fig. 7f).

## Discussion

The development of CRPC is a critical problem in patients with prostate cancer and there remains an urgent need to identify targets that do not function by activating AR to develop more effective therapies. Given that Wnt-11 is upregulated in CRPC and upon AR inhibition[28,29], components of the Wnt-11 signaling pathway could provide such targets. Here, we have identified FZD$_8$ is a major Wnt-11 receptor in prostate cancer that may be a useful therapeutic target.

Our findings show that FZD$_8$, like Wnt-11, is highly expressed in more aggressive prostate cancer cell lines. Expression of FZD8 mRNA is also elevated in tumor samples of prostate cancer datasets. We further show that both Wnt-11 and FZD$_8$ protein levels are higher in prostate tumor cells than in prostate epithelial cells in benign tissue, consistent with a role for FZD$_8$ in transducing autocrine Wnt-11 signals. We also observed a trend for increased expression of FZD$_8$ and Gleason score. That this was not significant might be attributed to the low number of high Gleason score samples in the TMA. Analysis of larger patient cohorts will be required to determine the prognostic value of FZD$_8$ in prostate cancer. In addition to the correlations between WNT11 and FZD8 and genes involved in EMT (Supplementary Fig. 6b), FZD8 expression strongly correlated with the presence of the TMPRSS2-ERG gene fusion and ERG mRNA expression (Supplementary Fig. 2d), an observation confirmed in a second dataset (Grasso; Oncomine$^{TM}$ database, Supplementary Fig. 2e). Further studies will be required to determine the relevance of this potentially interesting observation.

Studies in several cell types have found that Wnt-11 activates protein kinases, such as PKC and JNK[15], which can lead to activation of ATF/CREB family transcription factors[16] and inhibition of β-catenin/Tcf/LEF[19,20]. Wnt-11 activates ATF2 in a variety of cell contexts[16–18]. In prostate cancer cells, Wnt-11 and several FZDs activated ATF2-dependent transcription. Among them, FZD$_8$ and FZD$_{10}$ showed the strongest effects and also potentiated the effect of Wnt-11 and co-localized with Wnt-11. FZD10 mRNA was only detected in VCaP cells, which express the highest level of Wnt-11, suggesting that FZD$_{10}$ may be a functional Wnt-11 receptor in a subset of prostate tumors. However, FZD10 was not upregulated in prostate tumor datasets. In contrast, FZD8 was upregulated in prostate tumor datasets, correlated with WNT11 expression in prostate cancer cell lines and FZD$_8$ protein levels correlated with Wnt-11 in prostate TMAs. In addition, FZD$_8$ and Wnt-11 formed a stable complex in PC-3M cells. Together, these observations are consistent with FZD$_8$ as a receptor of Wnt-11 in metastatic prostate cancer. Wnt-11 is also upregulated upon hormone-depletion of LNCaP cells, where it is required for NED[28,29]. However, FZD8 expression was reduced in hormone-depleted LNCaP cells (Supplementary Fig. 1), suggesting other FZD family members transduce Wnt-11 signals in this context.

FZD8 silencing reduced Wnt-11 activation of ATF2-dependent and AP-1-dependent transcription, but did not affect β-catenin/Tcf/LEF-dependent gene reporter activity, consistent with Wnt-11/FZD$_8$ transducing a β-catenin-independent signal. FZD2 silencing also reduced Wnt-11 activation of ATF2-dependent transcription, albeit to a lesser extent than silencing of FZD8. However, FZD2 mRNA levels were not upregulated in prostate tumor datasets and FZD$_2$ did not co-localize with Wnt-11 in PC-3M cells. FZD4 mRNA levels were upregulated in prostate tumor datasets but FZD4 silencing did not affect Wnt-11 activation of ATF-2-dependent transcription and FZD$_4$ did not colocalize with Wnt-11, so FZD$_4$ is unlikely to transduce Wnt-11 signals in prostate cancer.

While this study focused on FZD class Wnt receptors, Wnt co-receptors are also anticipated to play a role in the response to Wnt-11. Several Wnt co-receptors were highly expressed in the majority of the prostate cancer lines examined, although none of them matched the WNT11 expression profile (Supplementary Fig. 1). Moreover, only PTK7 mRNA levels were upregulated in more than one prostate cancer dataset. Further studies will be needed to determine which Wnt co-receptors are important for Wnt-11 signaling in prostate cancer.

Previous studies have reported the involvement of Wnt-11 in prostate cancer migration and invasion[29]. Consistent with the role of FZD$_8$ as a Wnt-11 receptor, silencing of FDZ8 reduced prostate cancer cell migration and invasion. In keeping with the importance of Wnt signaling in tumor progression and EMT[41], FZD8 silencing reduced mesenchymal gene and protein levels, which may account for its requirement for prostate cancer cell migration and invasion.

A number of small molecule inhibitors have been developed that target Wnt signaling in cancer. The best known of these are porcupine inhibitors, which block Wnt secretion, tankyrase inhibitors, which inhibit β-catenin-dependent Wnt signaling by stabilizing Axin, and drugs that target β-catenin interactions with transcription factors[8]. The elucidation of the structure of the XWnt8-Fzd$_8$ CRD complex[55] has accelerated the development of drugs targeting the Wnt pathway at the receptor level[56,57]. The increased expression of FZD$_8$ in a significant proportion of prostate tumors and the demonstrated inhibitory effect of FZD8 silencing on prostate cancer cell migration and invasion suggests that inhibition of WNT-FZD$_8$ interactions may be a

useful approach for treatment of patients with metastatic prostate cancer. Consistent with this, we found that small molecule inhibitors that target the WNT-binding site on $FZD_8$[40] reduced prostate cancer cell migration.

Crosstalk between TGF-β and Wnt signals has been studied extensively[48]. The $FZD_8$ requirement for expression of EMT-related genes prompted us to explore its role in TGF-β signaling, finding that it is required for TGF-β effects on Smad phosphorylation, Smad-dependent gene reporter activity, expression of the TGF-β target genes PAI1 and MMP9, TGF-β-dependent cell invasion, and of a subset of TGF-β-regulated mesenchymal genes. While TGF-β increased MMP9 and PAI1 expression, it did not affect expression of EMT-related genes in PC-3M cells, apart from increasing CDH2 (Supplementary Fig. 10b), in contrast to what was observed in DU145 cells. This is consistent with the more mesenchymal and invasive character of PC-3M cells, as compared to DU145 cells[43]. Of particular interest, we observed crosstalk between TGF-β signaling and Wnt-11/ATF2 signaling. Dominant-negative ATF2 reduced TGF-β/Smad gene reporter activity and TGF-β-induced expression of mesenchymal genes, indicating a requirement for ATF2 in the TGF-β regulation of EMT genes. However, we were unable to confirm the previously reported interaction between ATF2 and Smad proteins. The requirement for $FZD_8$ in TGF-β signaling in prostate cancer is reminiscent of studies in lung fibroblasts, where TGF-β-induction of WNT5A, WNT5B, and FZD8 is required for the expression of genes encoding extracellular matrix proteins and myofibroblast differentiation markers[58].

As far as we are aware, this is the first report of an association between a member of the FZD family and the TGF-β receptor complex. The interaction is mediated by the $FZD_8$ CRD and the extracellular domain of TGFβRI and is affected by the presence of exogenous TGF-β. The molecular events that take place subsequent to TGF-β binding are complex[50]: in the absence of ligand, TGFβRI and TGFβRII may occur as monomers, homodimers, and heterodimers. TGFβRII homodimers and heterodimers are stabilized by contacts between the receptor cytoplasmic domains, whereas TGFβRI homodimers do not require the TGFβRI cytoplasmic domain. TGF-β binding to TGFβRII homodimers leads to recruitment of TGFβRI homodimers and formation of a heterohexamer complex of TGF-β, TGFβRI, and TGFβRII dimers. TGFβRII then phosphorylates and activates TGFβRI, which propagates the signal. Treatment of PC-3M cells with TGF-β for 30 min reduced the association between $FZD_8$ and TGFβRII without significantly affecting $FZD_8$ binding to TGFβRI (Fig. 7e), suggesting that $FZD_8$ plays a role in signal transduction subsequent to TGFβRII phosphorylation and activation of TGFβRI. However, the exact events that take place in the FZD8–TGFβR complex will require detailed study of the endogenous receptors.

While this manuscript was in preparation for submission, Li et al. reported that FZD8 promotes bone metastasis in prostate cancer[59]. The authors' observations that FZD8 is upregulated in prostate cancer and promotes prostate cancer cell migration and invasion are consistent with ours. However, they propose that FZD8 activates Wnt/β-catenin signaling by increasing WNT3A expression. In contrast, our results indicate that Wnt/β-catenin signaling activity is low in prostate cancer and is not affected by FZD8 silencing. How these contrasting mechanisms can be consolidated will require further studies.

In summary, our results suggest that by interacting with TGFβRI, $FZD_8$ can coordinate Wnt and TGF-β signals to promote expression of EMT genes and increase prostate cancer cell migration and invasion. $FZD_8$ may therefore be a useful therapeutic target in metastatic prostate cancer, since blocking its activity has the potential to inhibit aberrant activation of both Wnt and TGF-β signals.

## Methods

**Cell culture and reagents**. PC-3, LNCaP, and VCaP cells were obtained from the American Type Culture Collection. PC-3M cells were provided by Scott Fraser and Mustafa Djamgoz (Imperial College London), LNCaP and C4-2B cells were from Charlotte Bevan (Imperial College London), and DU145 cells were from John Masters (University College London) and Magali Williamson (Kings College London). Cell lines were authenticated by DNA profiling (Eurofins Genomics, Germany) and cells were routinely tested for mycoplasma. Low passage stocks were cultured for up to 6 months after thawing. LNCaP, C4-2B, PC-3M, and DU145 cells were cultured at 37 °C, 5% $CO_2$ in RPMI-1640 with GlutaMAX™ (Invitrogen) supplemented with 10% fetal bovine serum (FBS; First Link, UK) and antibiotics (100 units mL$^{-1}$ penicillin, 100 μg ml$^{-1}$ streptomycin, Invitrogen, UK). CSS treatment of LNCaP cells was by culture in RPMI with 5% charcoal-stripped serum (First Link, UK) for 3 or 6 days. VCaP cells were cultured in DMEM/Ham's F12, 10% FBS, antibiotics, 0.5% sodium pyruvate, and 1.5 mM glutamine. For transfection, cells were cultured in OptiMEM (Invitrogen). For gene reporter assays, TGF-β1 (R&D Systems) at 0.1 or 1 ng ml$^{-1}$ was added 4 h after transfection and incubated for 24 h. For q-RT-PCR, cells were treated with 1 ng ml$^{-1}$ TGF-β for 24 h. For invasion assays, cells were pre-treated with 5 ng ml$^{-1}$ TGF-β for 24 h and the same concentration was added when cells were plated on the transwell filters for 48 h. For evaluation of TGF-β-induced SMAD phosphorylation, 1 ng ml$^{-1}$ TGF-β was added for 30 min to cells 48 h after transfection. Inhibitor 1 (3235-0367) and 2 (2124-0331), which were obtained from ChemDiv Inc. (CA, USA) were used at 10 μM for 24 h in migration assays.

**Plasmids and siRNAs**. Plasmids used were pcDNA Wnt-11[29] and PA-tagged Wnt-11[60], a gift from Junichi Takagi, pRL-tk (Promega), Super8XTOP/FOP-Flash[11], AP-1-luciferase[61], and CAGA12-luc[62], which contains 12 CAGA boxes from the PAI1 promoter. ATF2-luciferase[63] was kindly provided by Christof Niehrs (Mainz, Germany) and pRK5 mFzd1-10-1D4[64] were from Chris Garcia and Jeremy Nathans (Addgene #42263-42272). CMV500 Δ-ATF2 and its empty vector CMV500[65] were from Charles Vinson (Addgene #33362, #33348), p3TP-lux[66] was from Joan Massagué and Jeff Wrana (Addgene #11767), pRK5 TGFβR1-Flag[67] and pRK5-Flag-ALK5-HA (ecto) were from Rik Derynck (Addgene #14831, Addgene #31720), pCMV5B TGFβR2 wild-type (N-terminal HA) was from Jeff Wrana (Addgene #24801), pRK5 mFz8CRD IgG (Addgene #16689)[68] and pCS2 LRP6N-IgG (Addgene #27279) were from Xi He. SMARTpool siRNAs were purchased from Dharmacon, ThermoFisher and dsiRNA from Integrative DNA Technologies (IDT, Leuven Belgium). The siRNAs and dsiRNA are listed in Supplementary Table 1.

**Cell transfection**. For gene reporter assays, 100,000 (C4-2B) or 80,000 (PC-3M and DU145) cells per well were plated in 12-well plates. After 24 h, cells were washed with OptiMEM and transfected with reporter constructs using Lipofectamine LTX with PLUS (Life Technologies), as instructed by the manufacturer. For silencing experiments, 60,000 PC-3M or DU145 cells per well were plated in 12-well plates. After 24 h, cells were washed with OptiMEM (Life Technologies) to remove antibiotics and transfected with 25 or 50 nM of siRNA or dsiRNA using RNAimax (Invitrogen), according to the manufacturer's directions. Silencing was evaluated after 48 h. In both cases, fresh media were added to the cells 4 h after transfection.

**Generation of stable knockdown cells**. For stable knockdown of FZD8, lentiviral pLKO and shFZD8 constructs (TRC Lentiviral Human shRNA, RHS4533, Dharmacon) were used. Lentiviruses were produced as previously described[69]. Lentiviral infection of PC-3M cells was performed twice by addition of cell media from transfected 293FT cells at 24 and 48 h. FZD8 mRNA levels were checked by q-PCR.

**RNA extraction and quantitative real-time PCR**. Total cellular RNA from prostate cancer cells was extracted using TRIzol (Invitrogen Life Technologies, Burlington, ON), according to the manufacturer's protocol. For the silencing experiments, RNA was extracted using illustra™ RNAspin Mini Isolation Kit (GE Healthcare). For RNA extraction from tumors growing on the chick CAM, tissues were broken up with a tissue homogenizer (Qiagen) for a few seconds and then lysed in 600 μl of RNA lysis buffer and RNA extracted using illustra™ RNAspin Mini Isolation Kit (GE Healthcare). In all cases, 2 μg total RNA was used for reverse transcription using M-MLV Reverse Transcriptase and RNase OUT Ribonuclease Inhibitor (Invitrogen), according to manufacturer's instructions. Quantitative-PCR was performed using PerfeCTa SYBR® Green Supermix, Low Rox (Quanta, Barcelona, Spain) in a Viia7 Real-Time PCR System (Applied Biosystems, Madrid, Spain) with the following conditions: Taq polymerase activation 95 °C 3 min, denaturation 95 °C 15 s, annealing/extension 62 °C 1 min, melting curve 95 °C 15 s, 60 °C 1 min, 95 °C 15 s, 40 cycles. Relative levels of mRNA were determined according to the ΔΔCT method, relative to the housekeeping gene 36B4. Primers are listed in Supplementary Table 2.

**Gene reporter assays**. Transfections were normally carried in triplicate wells 24 h after plating. Cells were transfected with 250 ng of ATF2/AP-1/TOP/FOP luciferase reporters, 50 ng pRL-tk, and 200 ng empty vector or Wnt-11 plasmid. For CAGA

luciferase analysis, cells were transfected with 450 ng of reporter and 50 ng pRL-tk. When FZD plasmids were co-transfected, 150 ng of reporter was used together with 50/100 ng of FZD plasmid. In experiments involving gene silencing, reporters and other plasmids were transfected 24 h after siRNAs. 24 h after transfection of reporters, cells were washed twice with PBS and lysed in Passive Lysis Buffer (Promega). Luciferase activity was measured using the Dual Glo Luciferase Assay System (Promega) or Luciferase Assay Kit (PJK, Germany) as instructed by manufacters. Gene reporter activities were calculated as luciferase/renilla ratios.

**Cell migration and invasion assays.** 250,000 PC-3M or DU145 cells per well were plated in 6-well plates and transfected with FZD8 siRNA for 48 h. Afterwards, cells were trypsinized and resuspended in RPMI with 1% FBS, and 50,000 cells/well were added to duplicate uncoated or Matrigel-coated 8 μm pore transwell filters with a polycarbonate membrane (Corning) for migration and invasion assays, respectively. When the effects of inhibitors 1 and 2 (I1, I2) were tested, DMSO or inhibitors were added at 10 μM to 50,000 cells/well on uncoated 8 μm pore transwell filters for migration assays. For TGF-β-dependent invasion assays, siRNA-transfected cells were cultured overnight in media with 0.2% FBS and then pretreated with 5 ng ml$^{-1}$ TGF-β for 24 h. Cells were then trypsinized and resuspended in media containing 0.2% FBS and 5 ng ml$^{-1}$ TGF-β. 50,000 cells/well were added to Matrigel-coated 8 μm pore transwell filters. In all cases, inserts were set in 24-well plates with media containing 20% FBS in the lower chamber. As a control for cell viability, cells were plated in parallel at the same density in 24-well plates. Migration and invasion were evaluated after 24 and 48 h, respectively. Non-migrating/invading cells were removed using a cotton swab and migrating/invading cells were stained using 0.1% crystal violet, 20% methanol, 0.36% PFA in PBS. Pictures were taken of five different fields with a 10× objective and the average numbers of migrating/invading cells per insert determined by counting stained cells. The control plate was also stained using crystal violet and absorbance measured at 595 nm after solubilizing in acetic acid. These data were then used to normalize the numbers of migrating/invading cells to the numbers of viable cells.

**Protein extraction and western blotting.** For total cell extracts, cells were lysed for 10 min in RIPA (50 mM Tris-HCl, pH 7.4, 150 mM NaCl, 0.25% deoxycholic acid, 1% NP-40, 1 mM EDTA (Millipore)) with cOmplete™ EDTA-free Protease Inhibitor Cocktail (Roche), PhosSTOP Phosphatase Inhibitor Cocktail Tablets (Roche), and 0.1% SDS (Life Technologies) and then centrifuged for 10 min at 15,000 × g. Extracts were separated on SDS polyacrylamide gels using a Mini Protean System (BioRad). After blocking with 3% BSA in TBS-T (Tris buffered saline, 0.1% Tween 20), blots were incubated with primary antibodies (Supplementary Table 3) at 4 °C overnight. After washing, blots were incubated for 1 h in blocking buffer with HRP-conjugated secondary antibodies diluted 1:20,000 (Jackson ImmunoResearch). Membranes were developed using chemiluminescence (Amersham ECL Western Blotting Detection Reagents, GE Healthcare). All unscropped scans of Western Blot are included in Supplementary Fig. 12.

**Immunoprecipitation.** 500,000 PC-3M cells were plated in 6-cm plates and transfected with 2.5 μg of total DNA (1:2 ratios for FZD8-1D4:Wnt-11-PA, FZD8-1D4:TGFβRI-Flag/HA-TGFβRII, FZD8-CRD-IgG:TGFβRI/II, and FZD8-CRD-IgG:Flag-ALK5-HA(ecto) and 5:10:1 ratio for Flag-SMAD3:HA-ATF2:HA-JUN). After 24 h, cells were washed twice with PBS and lysed in 1 ml of IP lysis buffer (50 mM Tris-HCl, pH 8, 150 mM NaCl, 1% Triton X-100, 1 mM EDTA (Millipore) with cOmplete™ EDTA-free Protease Inhibitor Cocktail (Roche) and PhosSTOP Phosphatase Inhibitor Cocktail Tablets (Roche)) for 5 min on ice. Samples were then centrifuged at 4 °C for 12 min at 15,000 × g. For inputs, 50 μl of supernatant was combined with 2× Laemmli Buffer (Sigma) and heated for 10 min at 37 °C (this temperature was used to reduce Fzd aggregation) or stored at −80 °C. For IPs, supernatants were incubated for 90 min on ice with 1 μg of antibody (anti-PA, anti-1D4, or anti-HA). Protein A/G PLUS-Agarose (Santa Cruz) was added to each sample and incubated for 1 h on a rotating wheel at 4 °C. For fusion proteins, IP was performed without antibody incubation. IPs were then collected by centrifugation at 500 × g for 30 s. Pellets were washed three times with IP buffer and once with ice-cold TBS. After washes, samples were re-suspended in 15 μl of 2× Laemmli Buffer and heated for 10 min at 37 °C. Primary antibodies used for blotting are described in Supplementary Table 3.

**Immunofluorescence.** PC-3M cells were plated on coverslips at 40,000 cells per well in a 24-well plate. 24 h after plating, cells were transfected with 170 ng Wnt-11 and 80 ng FZD plasmid or 120 ng TGFβRI/II and 90 ng FZD8-1D4. The following day, cells were fixed in 4% paraformaldehyde in PBS (Santa Cruz) for 20 min at RT. Permeabilization was performed using 0.1% Triton X-100 in PBS for 10 min at RT. Blocking was performed using 2% BSA, 50 mM glycine, 0.01% NaN₃ in PBS for 40 min to 1 h. Primary antibodies to Wnt-11, Fzd-8 and 1D4, HA and Flag epitope tags (Supplementary Table 3) were added overnight at 4 °C. Following incubation with secondary antibodies at 1:500 (AlexaFluor488, AlexaFluor594, Life Technologies), coverslips were mounted using Vectashield mounting medium with DAPI (Vector Labs). Stained cells were visualized using a confocal microscope (Leica TCS SP2) with a 63× oil immersion objective. For analysis of colocalization, 10 cells

from two independent experiments were quantified using ImageJ and the Coloc 2 plug-in and Pearson's correlation coefficient determined.

**Miniaturized 3D culture and imaging.** For miniaturized 3D cultures, PC-3 or PC-3M cells were plated at 150,000 cells/well in 6-well plates and transfected 24 h later with control or FZD8 siRNAs. After 48 h, 3D cultures were set up by embedding silenced cells between two layers of Matrigel™ and culturing them for 8–9 days on uncoated 96-well Angiogenesis μ-plates (Ibidi GmbH, Martinsried, Germany). This was done by adding 10 μl of Matrigel™/culture medium (1:1) per well, allowing to polymerize at 37 °C for 1 h and then adding 650 cells in 20 μl Matrigel™/culture medium (1:4) and incubating at 37 °C for 4 h to polymerize. Finally 60 μl of medium was added on top. Humidity was maintained by adding PBS to surrounding wells. Twelve replicate wells were used for each condition and media were changed every second day. Live cell imaging of the 3D assay plates was performed using an IncuCyte® system (Essen BioScience, Hertfordshire, UK). Images were captured every 2 h for 8–9 days. To monitor silencing efficiency during 3D culture, siRNA-transfected cells were replated in 12-well plates and RNA extracted at days 4 and 8 and analyzed for gene expression using q-RT-PCR.

**3D image acquisition and morphometric analyses.** At the experimental end-points, 3D multicellular structures were stained using 1 μM Calcein AM (Invitrogen) and ethidium homodimer-1(Invitrogen) in media at 37 °C for 30 min. Confocal images of four different fields per well were taken using a Zeiss spinning disk confocal unit with a 5× objective. Intensity projections were created with SlideBook (Intelligent Imaging Innovations Inc., Denver, CO, USA) and background noise was removed by normalization, also using SlideBook. Finally, confocal images were analyzed with VTT AMIDA software[70] and raw numerical data were statistically processed and visualized with R/Bioconductor. Morphometric parameters used in the analysis can be found in Supplementary Table 4; a more extended explanation of the parameters has been described previously[70].

**Tumor growth assay on the CAM.** Fertilized white Leghorn chicken eggs were cleaned with water and 70% ethanol. Eggs were placed into an egg incubator with the sharp end down (embryo development day 0, EDD0). Incubation was performed at 37 °C under constant humidity (60%) and rotation. No method of randomization was used. Separation of the developing CAM was induced on EDD4 by cutting a 2 mm diameter hole at the sharp end. After covering holes with tape, eggs were returned to the incubator. At EDD7, holes were enlarged to a final diameter of approximately 1 cm and a plastic ring was set above the blood vessels of the CAM. 500,000 PC3 cells/egg (previously transfected with siRNAs for 48 h) were suspended in PBS and Matrigel (1:1) and 20 μl cells per egg implanted in the middle of the ring. At EDD10, eggs were placed on ice for 1 h to anesthetize the embryos. Holes were enlarged and tumors were photographed in ovo and excised. Tumor area was measured blind using ImageJ and photographs from three independent experiments, each with 15 eggs per condition, were taken. Tumor specimens were excised from the CAM and placed in 4% paraformaldehyde for fixation. After that, tumors were paraffin-embedded and cut in 5-μm sections for immunohistochemical analysis.

**Clinical samples.** TMAs were generated at the Imperial College Experimental Cancer Medicine Centre using samples provided by the Imperial College Healthcare NHS Trust Tissue Bank (ICHTB; project R15043), which is supported by the National Institute for Health Research (NIHR) Biomedical Research Centre, based at Imperial College Healthcare NHS Trust and Imperial College London. Tissues were obtained from surgical resections from 99 prostate cancer patients following patient consent and approval from the local research ethics committee (ref: ICHTB HTA; licence: 12275; REC Wales approval: 12/WA/0196). The clinical characteristics of the patients are summarized in Supplementary Table 5. Other investigators may have received samples from these same tissues. TMAs contained two cores from regions containing cancer and two cores from regions without cancer from each patient. A histopathologist (J.C.) examined representative haematoxylin and eosin-stained sections to evaluate their pathology. Nine cores did not contain cancer and three showed low-quality staining or loss of sample and were not analyzed further. The staining intensities of FZD₈ and Wnt-11 were scored independently by two people (V.M.G. and R.M.K.) and divergences in scores were re-evaluated until consent was found. Each core was scored based on the staining intensity as 0 (no staining), 1 (weak staining), 2 (moderate staining), or 3 (strong staining).

**Immunohistochemistry.** Paraffin blocks containing lymph node metastases surgically removed from mice 31 days after orthotopic implantation of 2 × 10⁶ PC-3M-luc cells were provided by Genscript (Piscataway, NJ, USA). Sections from these blocks, from CAM tumors, and from TMAs were de-paraffinized with Histo-Clear II (National Diagnosis) and then transferred through four changes of 100, 96, 70% ethanol and water. Antigen retrieval was performed in a pressure cooker filled with sodium citrate buffer at pH 6.0. Endogenous peroxidase activity was quenched for 10 min with 3% hydrogen peroxide. Blocking was performed for 15 min with Avidin followed by 15 min with Biotin (Avidin/Biotin blocking kit; Vector Labs). Samples were washed with PBS and blocked with 5% horse serum for 30 min at room temperature to reduce non-specific staining. After washing, primary antibodies to

FZD$_8$, Wnt-11, Vimentin, and pan-Cytokeratin (CK) (Supplementary Table 3) were applied overnight at 4 °C. Sections were incubated with biotinylated secondary antibody (Vector Labs) for 30 min followed by Vectastain® Elite ABC reagent (Vector Labs) for 30 min. Liquid diaminobenzidine (DAB) (DAKO) was used as a chromogenic agent for 1–2 min and sections were counterstained with Mayer's haematoxylin. Images were taken on an AxioImager D1 light microscope (Zeiss).

**Statistical analysis**. Results are presented as the mean ± standard deviation (SD). All experiments were repeated at least three times. Statistical evaluations were performed with GraphPad Prism 5.0 (GraphPad, La Jolla, CA, USA) using two-sided Student's $t$-test for single comparisons or one-way analysis of variance (ANOVA) with post hoc Tukey for multiple group comparisons. A two-tailed $p$-value ≤ 0.05 was considered to indicate statistical significance. For TMA analysis, patients were divided into low (0, 1) and high (2, 3) FZD$_8$ and Wnt-11 expression and Gleason scores ≥4 + 3 and ≤3 + 4 and analyzed by one-way v2 test, Chi-squared test with Yates correction or Fisher's exact test, two-sided. Correlation analysis was calculated using Phi-correlations to test association between expression of FZD$_8$ and Wnt-11. All TMA analyses were performed using SPSS v16 (IBM Corp., Somers, NY, USA).

**Data availability**. All data generated or analyzed during this study are included in this article or the Supplementary Information files, or available from the authors upon request.

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

## Acknowledgements

We thank Iskander Aurrekoetxea and Giacomo Domenici for assistance generating stable knockdown cells, Inmaculada López-Sánchez for support with correlation quantification, and Maria Vivanco for critical reading of the manuscript. We also thank Jenny Steele and Naina Patel for help with the TMAs, Ignacio Zabalza (Galdakao Hospital) for advice interpreting the immunohistochemistry, Arkaitz Carracedo for anti-human vimentin, Junichi Takagi (Osaka University) for PA-Wnt-11, Pedro Lazo (IBMCC) for HA-ATF2 and HA-c-JUN, Wei Cui for Flag-SMAD3, and Christof Niehrs (Mainz, Germany) for ATF2-luciferase. We gratefully acknowledge funding from the Basque Department of Education (BFI-2010-129 and PRE_2015_0076), the Ministry of Science and Innovation (MICINN SAF2014-51966-R, SAF2017-84092-R), EMBO (STF_7003), Harris Family Charitable Trust, Academy of Finland (Phenotypic Screening for Cancer Drug Discovery/Consortium, PESCADoR 309372), Sigrid Jusélius Foundation, Finnish Cancer Organizations and Magnus Ehrnrooth Foundation, and infrastructure support from the Department of Industry, Tourism and Trade (Elkartek) and Department of Innovation Technology of the Government of the Autonomous Community of the Basque Country, the Center of Excellence Severo Ochoa (2017-2021), the Cancer Research UK Imperial Centre, the Imperial Experimental Cancer Medicine Centre and the National Institute for Health Research Imperial Biomedical Research Centre.

## Author contributions

V.M.G.: Conception and design, collection and assembly of data, data analysis and interpretation, and writing of manuscript; I.G.E.: administrative and technical support, collection and assembly of data; M.Å.: collection and analysis of data of 3D culture assays; M.C.P.: technical performance, data analysis, and interpretation of CAM assays; L.S.: data analysis of CAM assays, financial support; M.N.: data interpretation of 3D culture assays and administrative support; J.C.: histopathology; J.W.: administrative and financial support; R.M.K.: conception and design, financial support, data analysis and interpretation, and writing of manuscript.
