## [Peer Review File · Nature Communications]

Reviewers' comments:

Reviewer #1 (Remarks to the Author):

In this study the authors show that FZD8 is the major Wnt-11 receptor in prostate cancer. They report that FZD8 mediates Wnt and TGF-beta signaling and promotes EMT in prostate cancer cells. Finally, they show that FZD8 binds throughout its cysteine binding domain to the extracellular domain of the TGF-beta receptor 1 at the cell membrane. While these molecular details are potentially relevant, the manuscript fails to provide strong evidence of the relevance of these findings in prostate cancer progression.

Overall, the data are limited and weak. The approaches used are very basic and the data appear at the best preliminary compared with the author's claims. The authors propose targeting FZD8 as a strategy to treat castration resistant prostate cancer. However, the authors did not perform any experiment in vivo and any testing to prove their points.

Specific comments

In showing the impact of FZD8 on the malignant phenotype, the authors show very limited in vitro data (wound- healing assay and Boyden chamber). There are no in vivo studies to show the impact of FZD8 on migration and metastasis. The authors should establish stable loss of function models to test the effects in vitro and in vivo. Similarly, regarding the connection with TGF-beta signaling the authors fail to provide clear data on its functional relevance. Many of the assays (promoter assays, gene expression) performed in the study show weak, limited effects. The analysis of prostate cancer samples is also incomplete. There are no data showing the impact of FZD8 and Wnt11 on prostate cancer progression. The prognostic association of FZD8 and Wnt11 should be explored. The correlation of FZD8 and Wnt11 expression is not sufficient to support the clinical relevance of the crosstalk. Limited data obtained by exploring the Oncomine database are shown. These basic analyses provide preliminary data that should be further supported by additional analyses. Additional bioinformatics data appear unrelated to the main findings. For examples, the connection with ERG is informative but there is no follow-up or link with FZD8 and Wnt11. This is a potential interesting point but it needs further studies to make any conclusion.

Reviewer #2 (Remarks to the Author):

Wnt11 has been shown to play a role in prostate cancer biology, especially in invasion and metastasis. In this study Murillo et al demonstrate, that in prostate cancer cells Wnt11 mainly signals through Fzd8 as its receptor. In addition, Fzd8 was shown to regulate Tgf-beta/Smad signaling. Overall, this study is thoroughly conducted, contains several lines of evidence concerning the proposed mechanistic interactions and is well written. A few concerns remain:

- Fig 1b contains pictures of co-transfections of Wnt11 and different Fzd receptors. It is necessary to have both channels represented separately plus the merged picture (like you do in Fig 6b) to be able to visualize the co-localization. In addition, a clear high resolution picture of a single cell for each case (e.g. as a zoom in) is needed. Co-localization should also be further confirmed with a program like Image J – Colocalization Analysis.

- Fig. 2b How do you explain, that all Fzd receptor co-transfections with Wnt11 resulted in an increase of ATF2 activity? This seems to mean that all Fzd receptors can transduce Wnt11.

Minor points:

- "u" as a weight unit should be replaced by "µ". Please correct.
- All over the manuscript the °C symbol is not used uniformly. Please correct.
- Do you have an explanation why AXIN2 expression increases after FZD8 silencing?

- In Figure 1a you highlight FZD4 without further discussing it in the text.
- Supp. Table 5: Is FZD1 also undetectable in VCaP?

Remarks concerning figures:

- o Figure 1b: Please add significance of *,** and *** in the legend.
- o Figure 2f: empty square with with "siFZD8" missing
- o Figure 3d: Significance stars poorly readable in some cases
- o Figure 4a: FZD8 subtitle of prostate carcinoma higher than the rest.
- o Figure 6f: Explain dashed arrow somewhere.
- o Supp. Figure 5c: Please correct Realtive to Relative.
- o Supp. Table 1: Please correct Tecnoologies to Technologies

IMPORTANT: Please include Supp. Figure with all uncut Western Blots.

Reviewer #3 (Remarks to the Author):

Murillo-Garzón et al. demonstrate that FRZ8 is the main receptor for Wnt-11 in prostate cancer cells. They provide substantial evidence for high FRZ8 and Wnt-11 in prostate carcinoma, relative to benign prostate. Furthermore, they provide compelling evidence for a novel interaction between FRZ8 and TGF-beta receptors that plays a role in regulating a transcription of a subset of TGF-beta genes. While these results are interesting and important, the functional relevance is less well developed.

The authors make the claim that FRZ8 regulates the transcription of EMT-related genes, which drives prostate cancer invasion. I think this is overstated and not fully supported by the data presented. FRZ8 seems to have a modest effect (<20%) on the transcription of a subset of EMT-related genes. There is currently some controversy about what precisely constitutes EMT, and there is growing evidence in other cancer types that EMT is not necessary for invasion. In the current study, the relevance of the modest regulation of EMT-genes is unknown. Moreover, the study is limited in that it only reports mRNA expression, not protein expression. Nor are the phenotype of the cells reported.

Specific comments:

1) Fig. 1b - The resolution of the images isn't high enough to determine colocalization. It looks like they are co-expressed in cells, but this is meaningless if the two proteins were exogenously expressed. Higher resolution with split channels would improve the figure. Also, the field is zoomed out, and some cells appear to express neither protein or only one. The fields should be at higher magnification with emphasis on double positive cells. Also, I question the relevance of the low cell density (i.e. no cell-cell contacts). In prostate cancers, the cells are touching. It's possible that cell density will influence the co-localization of proteins.

2) It's not clear what the relevance of the relationship of FRZ8 to TMPRSS2-ERG fusion and ERG mRNA expression is?.

3) Fig. 3d - The changes in mRNA are relatively modest (<20%). Are the protein levels changed? Is there a change in cell morphology?

Do these markers correlate with FRZ8 expression in the data sets analyzed in Figure 1? Eg. in the TCGA dataset?

The carcinoma evaluated in Fig 4 with high FZD8 and Wnt-11 don't look mesenchymal, but look like well-organized, more typical of an epithelial phenotype. But higher resolution images are necessary to really determine this. Are EMT markers at the protein-level upregulated in FRZ8-high invasive prostate carcinoma?

4) Fig 4e - How do we know what is PC-3M cell and what is lymph node? The cell morphologies are difficult to see and differentiate between the 3 samples in the images presented. Are all the cells PC-3M in the FRZ8 and Wnt-11 conditions? A marker that can confirm the presence of the cancer cells or distinguish them from the lymph node is necessary.

5) Fig 5a - siFZD8 does reduce the signal, but it is still much higher than in the non-treated samples, suggesting that FRZ8 is not the only, or even main mechanism downstream of TGF-beta in this context. This should be clarified in the text.

6) Fig 5b - it is not clear if the analysis of TGF-beta target genes was conducted in cells treated with TGF-beta. If so, this should be described in the figure legend.

7) Fig. 5c - The changes in protein levels are not dramatic and would benefit from quantification of the band intensities.

8) Fig. 6b - The IF image is not overly convincing to conclude they colocalize. They would be expected to colocalize at the plasma membrane, but both look cytoplasmic. Higher resolution images, with membrane markers and images with split channels, in addition to the overlay, are necessary.

NCOMMS-17-18508A

Frizzled-8 integrates Wnt-11 and TGF-beta Signaling via TGFBR1

Reviewers' comments and authors' responses

Reviewer #1 (Remarks to the Author):

In this study the authors show that FZD8 is the major Wnt-11 receptor in prostate cancer. They report that FZD8 mediates Wnt and TGF-beta signaling and promotes EMT in prostate cancer cells. Finally, they show that FZD8 binds throughout its cysteine binding domain to the extracellular domain of the TGF-beta receptor 1 at the cell membrane. While these molecular details are potentially relevant, the manuscript fails to provide strong evidence of the relevance of these findings in prostate cancer progression. Overall, the data are limited and weak. The approaches used are very basic and the data appear at the best preliminary compared with the author's claims. The authors propose targeting FZD8 as a strategy to treat castration resistant prostate cancer. However, the authors did not perform any experiment *in vivo* and any testing to prove their points.

We thank the reviewer for these comments. In the revised manuscript, we have strengthened the data by adding new experiments using additional approaches *in vitro* 3D and *in vivo* assays

Specific comments

In showing the impact of FZD8 on the malignant phenotype, the authors show very limited *in vitro* data (wound- healing assay and Boyden chamber). There are no *in vivo* studies to show the impact of FZD8 on migration and metastasis. The authors should establish stable loss of function models to test the effects *in vitro* and *in vivo*.

We now have confirmed the effects we observed using transient FZD8 silencing on prostate cancer cell invasion using cell lines in which FZD8 has been stably silenced using lentiviral constructs expressing FZD8 shRNAs. The results are shown in Supplementary Figure 5E-G. In addition, we have added the results of a comprehensive set of 3D cell culture experiments that further support a role of FZD8 in invasion (Figure 4 and Supplementary Figure 7) and we have addressed the role of FZD8 *in vivo* using the chick chorioallantoic membrane (CAM) assays (Figure 5). Together, these results complement the *in vivo* results from the recent study by Li *et al*, who showed that FZD8 silencing inhibits prostate cancer metastasis (Cancer Letters, Aug 28 2017;402:166-176).

Similarly, regarding the connection with TGF-beta signaling the authors fail to provide clear data on its functional relevance. Many of the assays (promoter assays, gene expression) performed in the study show weak, limited effects.

We have now examined the functional relevance of the connection with TGF-beta signaling by showing that FZD8 silencing inhibits TGF-beta-induced prostate cancer cell invasion. The results are shown in Figure 7D and Supplementary Figure 9e-g.

The analysis of prostate cancer samples is also incomplete. There are no data showing the impact of FZD8 and Wnt11 on prostate cancer progression. The prognostic association of FZD8 and Wnt11 should be explored. The correlation of FZD8 and Wnt11 expression is not sufficient to support the clinical relevance of the crosstalk.

None of the patients that provided material for the TMA have died from prostate cancer and there were no differences in FZD8 or Wnt-11 levels in patients with or without rising PSA

after surgery in this cohort. We have therefore explored the potential prognostic association of FZD8 and WNT11 using publically available gene expression data, using the Taylor dataset. The results show that recurrence is more frequent in patients with tumors expressing elevated levels of both FZD8 and WNT11, compared to patients in which the expression of neither gene or expression of only one gene is elevated. The results are shown in Supplementary Figures 8b and 8c.

Limited data obtained by exploring the Oncomine database are shown. These basic analyses provide preliminary data that should be further supported by additional analyses.

We have expanded the bioinformatics analysis and the results that indicate increased expression of FZD8 and WNT11 in tumors with lymph node invasion, shown in Supplementary Figure 8a.

Additional bioinformatics data appear unrelated to the main findings. For examples, the connection with ERG is informative but there is no follow-up or link with FZD8 and Wnt11. This is a potential interesting point but it needs further studies to make any conclusion.

We agree that follow-up experiments will be required to explore the link with ERG further and feel that these go beyond the scope of this manuscript and have moved the data from the Results section to the discussion as this is a potentially interesting observation.

Reviewer #2 (Remarks to the Author):

Wnt11 has been shown to play a role in prostate cancer biology, especially in invasion and metastasis. In this study Murillo et al demonstrate, that in prostate cancer cells Wnt11 mainly signals through Fzd8 as its receptor. In addition, Fzd8 was shown to regulate Tgf-beta/Smad signaling. Overall, this study is thoroughly conducted, contains several lines of evidence concerning the proposed mechanistic interactions and is well written.

We thank the reviewer for these positive comments.

A few concerns remain: - Fig 1b contains pictures of co-transfections of Wnt11 and different Fzd receptors. It is necessary to have both channels represented separately plus the merged picture (like you do in Fig 6b) to be able to visualize the co-localization. In addition, a clear high-resolution picture of a single cell for each case (e.g. as a zoom in) is needed. Co-localization should also be further confirmed with a program like Image J – Colocalization Analysis.

These experiments have been repeated and analyzed by confocal microscopy and analyzed using ImageJ. The new images are shown in Figure 1b and Supplementary Figure 2 and the results of the colocalization analysis are shown in Figure 1b.

- Fig. 2b How do you explain, that all Fzd receptor co-transfections with Wnt11 resulted in an increase of ATF2 activity? This seems to mean that all Fzd receptors can transduce Wnt11.

Previous studies have shown that some Fzd family members activate of beta-catenin-dependent signaling when transfected alone. It has been proposed that these Fzds either potentiate the activity of endogenous Wnt ligands or activate signals in a ligand-independent, context-dependent manner (for example, see Umbhauer *et al.*, EMBO Journal, 2000; Carron *et al.*, J Cell Science, 2003). In Fig 2b, the black bars indicate that transfection of most Fzd family receptors alone is sufficient to increase ATF2 activity, so this may also be the case for ATF2. The white bars show activity when Wnt-11 is co-transfected and while they show higher activity than the black bars in many cases, this is because transfection of Wnt-11 alone

also activates ATF2. The key comparison is the levels of the white bars versus empty vector (pRK5), where the only significant increases observed are for Fzd8 and Fzd10.

Minor points:

- "u" as a weight unit should be replaced by "μ". Please correct. Corrected
- All over the manuscript the °C symbol is not used uniformly. Please correct. Corrected
- Do you have an explication why AXIN2 expression increases after FZD8 silencing? This could be related to de-repression of basal canonical Wnt signaling, as Wnt-11 has been found to inhibit canonical Wnt signaling (discussed in Uysal-Onganer and Kypta, Acta Physiologica, 2011).
- In Figure 1a you highlight FZD4 without further discussing it in the text A sentence has been added
- Supp. Table 5: Is FZD1 also undetectable in VCaP? Yes, information added to table

Remarks concerning figures:

- o Figure 1b: Please add significance of *, ** and *** in the legend.
- o Figure 2f: empty square with with "siFZD8" missing
- o Figure 3d: Significance stars poorly readable in some cases
- o Figure 4a: FZD8 subtitle of prostate carcinoma higher than the rest.
- o Figure 6f: Explain dashed arrow somewhere.
- o Supp. Figure 5c: Please correct Realtive to Relative.
- o Supp. Table 1: Please correct Tecnologies to Technologies The changes requested above have now been made.

IMPORTANT: Please include Supp. Figure with all uncut Western Blots. This has now been done.

Reviewer #3 (Remarks to the Author):

Murillo-Garzón et al. demonstrate that FRZ8 is the main receptor for Wnt-11 in prostate cancer cells. They provide substantial evidence for high FRZ8 and Wnt-11 in prostate carcinoma, relative to benign prostate. Furthermore, they provide compelling evidence for a novel interaction between FRZ8 and TGF-beta receptors that plays a role in regulating a transcription of a subset of TGF-beta genes. While these results are interesting and important, the functional relevance is less well developed. The authors make the claim that FRZ8 regulates the transcription of EMT-related genes, which drives prostate cancer invasion. I think this is overstated and not fully supported by the data presented. FRZ8 seems to have a modest effect (<20%) on the transcription of a subset of EMT-related genes. There is currently some controversy about what precisely constitutes EMT, and there is growing evidence in other cancer types that EMT is not necessary for invasion. In the current study, the relevance of the modest regulation of EMT-genes is unknown. Moreover, the study is limited in that it only reports mRNA expression, not protein expression. Nor are the phenotype of the cells reported.

Specific comments:

1) Fig. 1b - The resolution of the images isn't high enough to determine colocalization. It looks like they are co-expressed in cells, but this is meaningless if the two proteins were exogenously expressed. Higher resolution with split channels would improve the figure. Also, the field is zoomed out, and some cells appear to express neither protein or only one.

The fields should be at higher magnification with emphasis on double positive cells. Also, I question the relevance of the low cell density (i.e. no cell-cell contacts). In prostate cancers, the cells are touching. It's possible that cell density will influence the co-localization of proteins.

We thank the reviewer for these comments and suggestions. These experiments have been repeated and analyzed by confocal microscopy and analyzed using ImageJ. The new images are shown in Figure 1b and Supplementary Figure 2 and the results of the colocalization analysis are shown in Figure 1b. The proteins were exogenously expressed because the antibodies available were not sufficiently sensitive to detect the endogenous proteins. However, we believe these results are meaningful because we co-transfected Wnt-11 with each of the ten FZD family members, all harboring the same epitope tag, and colocalization was much higher for FZD6, FZD8 and FZD10 (Pearson > 0.4) than for the other family members.

2) It's not clear what the relevance of the relationship of FRZ8 to TMPRSS2-ERG fusion and ERG mRNA expression is?

We agree that follow-up experiments are required to explore the link with ERG and we feel that these are beyond the scope of this manuscript. We have therefore taken the information out of the Results section. However, as this is potentially very interesting, we still mention it in the discussion.

3) Fig. 3d - The changes in mRNA are relatively modest (<20%). Are the protein levels changed? Is there a change in cell morphology? Do these markers correlate with FRZ8 expression in the data sets analyzed in Figure 1? Eg. in the TCGA dataset?

To confirm the mRNA data, we have now also examined the effects of FZD8 silencing on the protein levels of some of the gene products. The results indicate that FZD8 silencing reduces N-cadherin and vimentin protein levels (Figure 3e). In addition, vimentin levels are reduced in FZD8-silenced tumors grown *in vivo* on the chick CAM (Figure 5e). We have also now used 3D assays to show that FZD8 silencing affects PC3 and PC3M sphere morphogenesis (Figure 4 and Supplementary Figure 7). Finally, we have compared FZD8 gene expression with that of epithelial and mesenchymal markers and found positive correlations between FZD8 and WNT11 with the mesenchymal genes SNAI1, SNAI3, TWIST1 and TWIST2 and negative correlations with the epithelial genes CDH1 and CTNNB1 (Supplementary Figure 6b).

The carcinoma evaluated in Fig 4 with high FZD8 and Wnt-11 don't look mesenchymal, but look like well-organized, more typical of an epithelial phenotype. But higher resolution images are necessary to really determine this. Are EMT markers at the protein-level upregulated in FRZ8-high invasive prostate carcinoma?

In the revised manuscript, we have added results from another patient tumor, including higher resolution images that show an area with disseminated tumor cells that are positive for FZD8, Wnt-11 and epithelial cytokeratins (Figure 6b). We have kept the original image also (now Figure 6a) as this is a Gleason score 4 tumor showing a 'cribriform' pattern that is found in invasive prostate cancer and is a strong prognostic marker for distant metastasis and disease-specific death in patients with Gleason sum score 7 prostate cancer at radical prostatectomy (Kweldam et al., Mod Pathol. 2015). Although the tumor cells do not have a mesenchymal appearance, they are clearly disorganized and their lack of 'organization' has been confirmed by a pathologist. We have also stained for vimentin but in this patient cohort it was mainly expressed in tumor stroma and did not correlate with FZD8. Previous studies reported quite different patterns of expression of vimentin in prostate cancer (Zhang et al, Clin. Can. Res.

2009, Kolijn et al., Oncotarget 2015, Figiel et al., Human Pathology, 2017). We were unable to examine other EMT markers as there was insufficient material available.

4) Fig 4e - How do we know what is PC-3M cell and what is lymph node? The cell morphologies are difficult to see and differentiate between the 3 samples in the images presented. Are all the cells PC-3M in the FRZ8 and Wnt-11 conditions? A marker that can confirm the presence of the cancer cells or distinguish them from the lymph node is necessary.

Sections have now been stained for human vimentin to localize the PC-3M cells. The results are shown in Figure 6f.

5) Fig 5a - siFZD8 does reduce the signal, but it is still much higher than in the non-treated samples, suggesting that FRZ8 is not the only, or even main mechanism downstream of TGF-beta in this context. This should be clarified in the text.

A sentence has been added to clarify this in the text (now Figure 7a).

6) Fig 5b – it is not clear if the analysis of TGF-beta target genes was conducted in cells treated with TGF-beta. If so, this should be described in the figure legend.

This was not the case in this experiment (now Figure 7b), where we relied on autocrine TGF- β activity. The experiment was however done +/- TGF- β in DU145 cells (Figure 7e).

7) Fig. 5c – The changes in protein levels are not dramatic and would benefit from quantification of the band intensities.

This has now been done (now Figure 7c).

8) Fig. 6b - The IF image is not overly convincing to conclude they colocalize. They would be expected to colocalize at the plasma membrane, but both look cytoplasmic. Higher resolution images, with membrane markers and images with split channels, in addition to the overlay, are necessary.

The images have been replaced by high-resolution confocal images with split channels (now Figure 8b). Immunocytochemistry for N-cadherin as a plasma membrane marker was also carried out, and the result for FZD8 and TGFBR2 shown below. This is not included in the manuscript itself as the species of antibodies available precluded us carrying out all the staining combinations, and we feel that the new images in the revised manuscript do now support the case for membrane colocalization.

In conclusion, we thank the reviewers for their suggestions to improve the manuscript and hope that they will agree that the additional experiments outlined above make this study worthy of publication in Nature Communications.

REVIEWERS' COMMENTS:

Reviewer #1 (Remarks to the Author):

The authors have addressed the main concerns I raised

Reviewer #2 (Remarks to the Author):

The reviewers questions were answered and I have no more major concerns. Even so I did not ask for this in the first round of submission, I know realized that in the title there is no reference to cancer in general or prostate cancer specifically. I feel that this reference is important, as the study is based on prostate cancer cell lines and patient material/data. The title as it is would need confirmation of the findings in normal tissues or other cancer types.

Reviewer #3 (Remarks to the Author):

The authors have made substantial revisions and addressed my previous concerns. This represents a novel and important contribution to our understanding of Wnt and TGF-beta signaling related to prostate cancer.